# *In silico* assessment of arrhythmic risk following the implantation of engineered heart tissues in porcine hearts with varying infarct locations

Ricardo M. Rosales[1,2]*, Gonzalo R. Ríos-Muñoz[3,4,5], Ana María Sánchez de la Nava[4], María Eugenia Fernández-Santos[4,5], Javier Bermejo[4,5,6], Manuel Doblaré[1,2,7], Ana Mincholé[1,2,7], Esther Pueyo[1,2,7]

**1** Aragón Institute for Health Research (IISA), Zaragoza, Aragón, Spain, **2** Aragón Institute of Engineering Research (I3A), University of Zaragoza, Zaragoza, Aragón, Spain, **3** Bioengineering Department, Universidad Carlos III de Madrid, Leganés, Spain, **4** Department of Cardiology, Instituto de Investigación Sanitaria Gregorio Marañón (IiSGM), Hospital General Universitario Gregorio Marañón, Madrid, Spain, **5** Center for Biomedical Research in Cardiovascular Disease Network (CIBERCV), Madrid, Spain, **6** Departamento de Medicina, Universidad Complutense, Madrid, Spain, **7** CIBER-BBN, Instituto de Salud Carlos III, Madrid, Spain

* rrosales@unizar.es

## Abstract

Engineered heart tissues (EHTs) have shown promise in partially restoring ejection fraction after myocardial infarction (MI); however, their potential to introduce electrophysiological heterogeneities and promote arrhythmias remains underexplored. This study assessed the arrhythmogenic risk following immature EHT engraftment in infarcted ventricles using computational simulations that replicate preclinical protocols. EHT computational models were developed and integrated into nine validated porcine-specific biventricular models from pigs with left circumflex (LCx, n = 4) or left anterior descending (LAD, n = 5) MIs. Ventricular tachycardia (VT) susceptibility was evaluated using an S1–S2 stimulation protocol across varying pacing sites and coupling intervals, accounting for infarct characteristics, implantation site, conductivity, and the ventricular conduction system (CS). VT burden was quantified with a 0–1 inducibility score (IS). *In silico* reentrant activity qualitatively reproduced the arrhythmic patterns observed experimentally in porcine MI models. VT vulnerability was greater in LAD than in LCx infarcts, consistent with a larger infarct size. Inclusion of the CS modified VT burden by providing conduction shortcuts that either facilitated or suppressed reentry. Remuscularization directly on the MI region (IS = 0.49) heightened VT inducibility in dense, transmural scars (IS = 0.16), whereas lateral EHT implantation (IS = 0.35) reduced this risk with respect to direct implantation. In non-transmural scars, VT inducibility varied with the implantation site. Matching EHT conductivity to host myocardium lowered or contained arrhythmogenicity (LCx-IS: from 0.5 to 0.25; LAD-IS: stable at 0.57). These results highlight the latent arrhythmic

which permits unrestricted use, distribution, and reproduction in any medium, provided the original author and source are credited.

**Data availability statement:** The modeling and postprocessing software are publicly available at https://github.com/lino202/HeartModelling and https://github.com/lino202/simOmPP, while the simulation solver is provided at https://github.com/lino202/ELECTRA. The computational models and simulation configuration files for the pre-remuscularization cases are available at https://doi.org/10.5281/zenodo.17415592. Corresponding files for post-remuscularization cases, along with the employed OM data obtained during VT, can be found at https://doi.org/10.5281/zenodo.17435811.

**Funding:** RMR, GRRM, AMSN, MEFS, MD, and EP received financial support from the EU H2020 Program under G.A. 874827 (BRAV3). RMR, AM, and EP were supported by Agencia Estatal de Investigación - Ministerio de Ciencia e Innovación (Spain) through projects PID2022-140556OB-I00, TED2021-130459B-I00, and CNS2022-135899, by European Social Fund (EU) and Aragón Government through project LMP94_21 and BSICoS group T39_23R, and by the European Research Council under G.A. 638284. RMR, MD, and EP were supported by Agencia Estatal de Investigación - Ministerio de Ciencia e Innovación (Spain) through project CARDIOPRINT (PLEC2021-008127). GRRM was supported by Madrid Government (Comunidad de Madrid) under the Multiannual Agreement with UC3M (FLAMA-CM-UC3M), and through project MAGERIT-CM (TEC-2024/COM-44). The funders had no role in study design, data collection and analysis, decision to publish, or preparation of the manuscript.

**Competing interests:** The authors have declared that no competing interests exist.

risk of EHT-mediated remuscularization after MI, identifying infarct substrate, EHT conductivity, and implantation site as critical determinants, and emphasize the importance of incorporating the CS for accurate risk assessment.

## Author summary

Remuscularization with engineered heart tissue patches has shown preclinical potential to restore cardiac ejection fraction after myocardial infarction. However, the introduction of electrophysiological heterogeneities from these patches remains underexplored. To address this, we performed *in silico* investigations of post-infarction arrhythmogenesis and its progression after patch engraftment. Using porcine models with vessel-specific infarcts, we conducted an arrhythmia inducibility protocol. We first achieved good qualitative agreement between our *in silico* simulated and the experimentally observed arrhythmic patterns in pigs. Before patch engraftment, we found that larger infarcts increased ventricular tachycardia vulnerability. The inclusion of the ventricular conduction system modified arrhythmic outcomes, as electrical shortcuts could either facilitate or suppress reentry. Following remuscularization, highly transmural scars exhibited higher arrhythmicity, with lateral patch implantation mitigating it compared to direct placement. In contrast, for poorly transmural scars, the relationship between arrhythmia inducibility and patch location was unclear, highlighting subject-specific computational models as a tool for individualized risk prediction. When host-like patch conductivity was used, arrhythmia inducibility after engraftment was reduced or contained. Overall, our results: i) reveal the latent arrhythmogenicity associated with patch-mediated remuscularization of infarcted hearts; ii) identify infarction substrate, patch conductivity, and implantation site as key arrhythmogenic determinants; and iii) emphasize the importance of accurately modeling the cardiac conduction system.

## 1 Introduction

Cardiovascular diseases remain the leading cause of mortality worldwide, with ischemic heart disease being the primary contributor, accounting for 33% of cardiovascular-related deaths in women and 40% in men annually [1]. In recent years, mortality rates of ischemic heart disease have increased in low- and middle-income countries, placing a significant economic burden through direct healthcare expenditures and substantial indirect costs, including productivity losses and unpaid care efforts [1]. This burden is projected to escalate due to the aging population of Europe and other regions, highlighting the urgent need for effective therapeutic strategies to treat ischemic heart disease and its most prevalent manifestation, myocardial infarction (MI) [1].

MI occurs when prolonged ischemia, typically due to inadequate reperfusion, induces irreversible cardiomyocyte death and subsequent tissue injury. This loss

of cardiomyocytes initiates a coordinated inflammatory response that ultimately replaces the necrotic myocardium with fibrotic tissue devoid of active contractile properties, leading to impaired systolic and diastolic function [2]. The structural remodeling associated with MI increases the susceptibility to heart failure and sudden cardiac death, likely linked to the development of sustained ventricular tachycardia (VT). The extent of post-MI cardiomyopathy is strongly influenced by the location of the infarct. MIs with occlusion of the left anterior descending (LAD) coronary artery generally lead to larger infarcts, more pronounced remodeling, and greater functional decline compared to MIs with occlusion of the left circumflex (LCx) coronary artery [3].

Current therapeutic options to prevent severe post-MI deterioration of cardiac function include heart transplantation and left ventricular assist devices [4,5]. Although heart transplantation remains the definitive treatment for end-stage heart failure, its widespread application is hindered by a limited donor pool and significant post-operative challenges, such as primary graft dysfunction, immune rejection, and the need for lifelong immunosuppressive therapy [6]. Similarly, assist devices for the left ventricle (LV), commonly used as a bridge to transplantation, involve major surgical intervention and require prolonged anticoagulation, posing additional risks to patients [4,5].

A promising tissue engineering strategy under active investigation is the remuscularization of the infarct scar using human-induced pluripotent stem cell-derived cardiomyocytes (hiPSC-CMs) to restore LV ejection fraction [7–13]. Recent studies have shown that the implantation of engineered heart tissues (EHTs) containing hiPSC-CMs can improve the post-MI ejection fraction in athymic rats with LAD-induced infarctions [9,10]. However, hiPSC-CMs remain structurally and functionally immature compared to adult cardiomyocytes, even when cultured in biomimetic environments incorporating electromechanical stimulation and biochemical cues within bioprinted melt electrowriting scaffolds [10,14]. Electrophysiologically, hiPSC-CMs exhibit spontaneous automaticity, elevated diastolic membrane potentials, prolonged action potentials (APs), and reduced upstroke velocities [15], resulting in suboptimal intercellular coupling and extended AP durations (APDs) in EHTs relative to the native myocardium [10,14]. Consequently, EHT-based therapies can introduce additional electrophysiological heterogeneities, potentially increasing the risk of arrhythmogenesis following MI.

Over the past few decades, advances in biological data acquisition, computational power, and data storage have propelled the emergence of *in silico* modeling and simulation as a robust framework for investigating cardiac electrophysiology, particularly for therapy evaluation and diagnostic support [16–18]. Computational electrophysiology provides substantial advantages over traditional clinical studies, offering greater experimental flexibility while substantially reducing the time, cost, and resources required. While human-based models offer the highest clinical relevance, their development is often limited by the scarcity of high-resolution, multi-scale datasets obtained from the same individual. Acquiring matched anatomical and functional data in human cohorts is technically and ethically challenging, frequently necessitating the use of generic or population-based parameters [19,20]. In contrast, animal models provide highly controlled experimental conditions that facilitate the acquisition of subject-specific data, enabling the development of precisely validated models essential for investigating complex phenomena such as cardiac resynchronization therapy, MI, and ventricular arrhythmogenesis [20–23].

Domestic *Sus scrofa* pigs are considered the gold standard due to their close physiological similarity to humans, particularly in coronary anatomy, hemodynamics, and electromechanical properties [24]. Porcine computational models have been instrumental in characterizing the arrhythmogenic substrate following MI [23] and in elucidating mechanisms by which ionic current blockade produces early or delayed afterdepolarizations [25]. Regarding EHT-mediated epicardial remuscularization, recent studies by Yu et al. [17] and Fassina et al. [26] have employed human biventricular (BiV) and LV frameworks, respectively, to investigate the electrophysiological impact of this therapy. However, because most experimental feasibility studies for post-MI remuscularization are conducted in large animal models [7,12,27], there remains a critical need for precise, species-specific computational assessments to accurately interpret and translate these preclinical results.

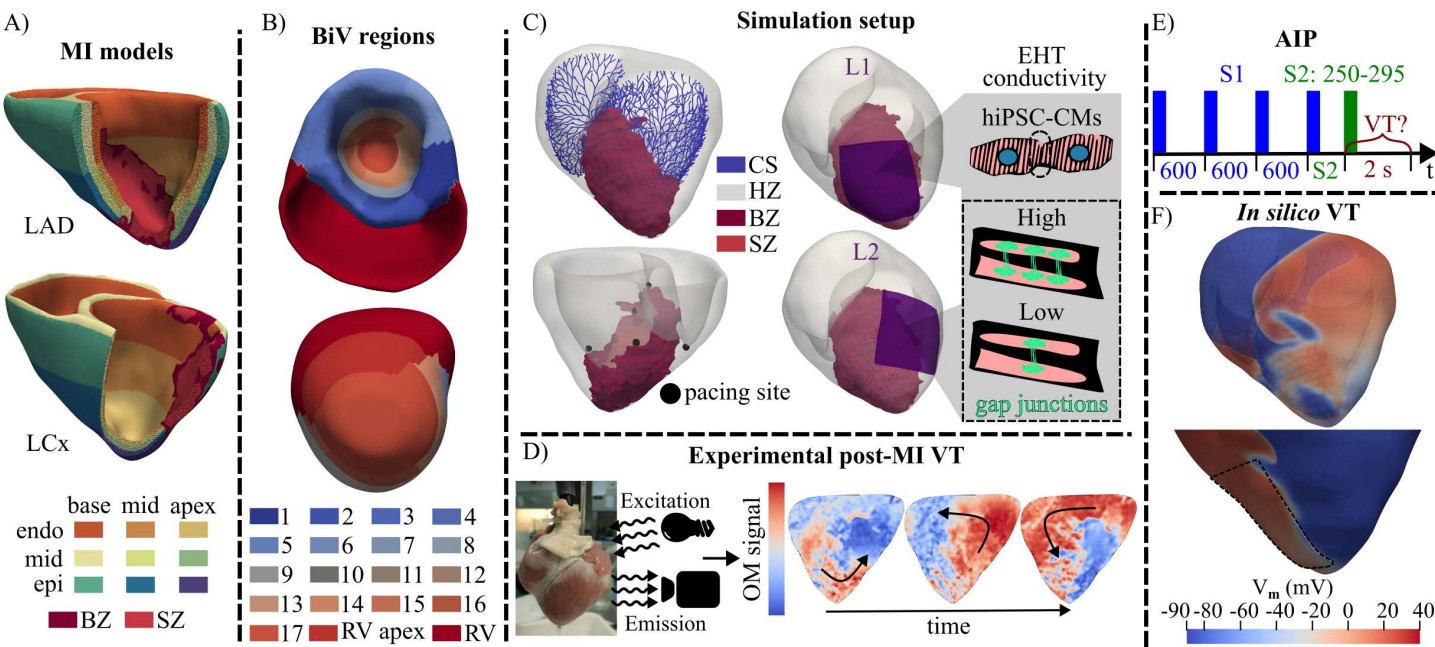

This study aims to elucidate the mechanisms driving post-MI arrhythmogenesis and how they are influenced by EHT engraftment. To this end, we performed *in silico* investigations of chronic post-MI arrhythmogenesis and its progression after the engraftment of EHTs in multiple MI models with varying infarct locations. We used our nine previously developed and validated, highly detailed porcine-specific MI models [20] and subjected them to an arrhythmia inducibility protocol (AIP). First, we qualitatively compared simulated arrhythmic dynamics with experimentally observed reentrant patterns obtained through optical mapping (OM). Next, we quantified VT burden across different infarct substrates, with and without the inclusion of the ventricular conduction system (CS), revealing key insights into the role of these factors in modulating arrhythmogenic potential. Furthermore, we performed a sensitivity analysis to determine how post-MI remuscularization with EHT influences arrhythmia susceptibility, thus identifying critical factors, such as EHT implantation site and electrical conductivity, that can exacerbate or mitigate arrhythmic risk in this therapeutic context. Building on our previous work [28–30], we expanded our modeling framework by incorporating high-resolution porcine-specific BiV models and EHT geometries, the use of a novel pig-specific cellular electrophysiology model, and evaluations related to the electrical maturation state and of the site of EHT implantation. We also applied an AIP in various MI morphologies and locations to robustly characterize arrhythmic risk.

## 2  Methods

Fig 1 shows the different steps of our proposed analysis. In summary, our nine previously developed and validated MI individualized models were used in an evaluation of arrhythmogenesis before and after remuscularization therapy. In line with our previous work, we refer to the four LCx samples as pigs 4–7 and to the five LAD samples as pigs 8–12 [20].

**Fig 1. Summary of the arrhythmic evaluation performed before and after EHT implantation.** A- Examples of BiV models of infarcts with LAD occlusion (top, pig 10) and LCx occlusion (bottom, pig 6). B- LV AHA segments and the apex of the right ventricle (RV) for another BiV model (pig 4). C- Example of a BiV model (pig 10) with its BZ, CS, pacing sites for the AIP and EHT implantation locations and maturation states. D- Experimental OM of the VT induced in pig 8. E- AIP depicting S1-S2 times. F- *In silico* VT induced in pig 10 before (top) and after (bottom) EHT (dashed, black line) remuscularization. All external elements are from public domain sources (Emitting light: https://openclipart.org/detail/185271/light-bulb-icon-2, Camera: https://openclipart.org/detail/343549/camera).

## 2.1 MI modeling

In a previous study, we built nine porcine-specific MI BiV models [20], two of which are shown in Fig 1A. Briefly, they consisted of four LCx and five LAD BiV models with infarcts that were delineated from late gadolinium-enhanced (LGE) cardiac magnetic resonance (CMR). LGE-CMRs were segmented using previously implemented automatic algorithms [31] with subsequent manual corrections. Based on the gadolinium washout and, therefore, the brightness of the LGE-CMR, porcine ventricles were divided into healthy (HZ), border (BZ), and scar (SZ) zones [20]. BZ represents the boundaries of MI, containing a mixture of viable tissue and collagen, while SZ defines the dense, non-conductive collagen remodeling site. Transmural heterogeneities were implemented by solving the steady-state diffusion (Laplace) equation with varying boundary conditions [32]. In accordance with established human studies [33,34], the endocardium, midmyocardium, and epicardium were defined using a 40:35:25 proportion. Additionally, an apicobasal gradient was defined by solving the Laplace equation with constant Dirichlet boundary conditions at the apex and base of the BiV domain. The BiV domain was subsequently partitioned into base, middle, and apex regions using an empirical proportion of 48:35:17 [20]. Standard fiber fields were generated with a rule-based model, as described by Bayer et al. [35]. The CS was generated for each model using a fractal tree algorithm and subsequently projected into the ventricular wall, following established methods [36,37]. The CS mesh was defined using line elements with a mean length of 528 µm, incorporating an average of 1623 Purkinje-muscular junctions (PMJs). CS-activated ventricular nodes were located within a maximum distance of 0.5 mm from the CS endpoints, with an average of 14.5 connections between each endpoint and the myocardial nodes to avoid source-sink mismatch-mediated conduction blocks at PMJs [20]. Detailed descriptions of the BiV MI and CS model construction and their validation against porcine-specific experimental data are provided in our previous work [20].

## 2.2 Arrhythmia inducibility

**2.2.1 Experimental protocol.** As can be observed in Fig 1D, induced arrhythmias in pigs 8–12 were experimentally measured by OM. The experimental preparation and the OM setup are described in detail in [20,38]. Briefly, the heart of LAD pigs was excised and placed on a Langendorff apparatus for OM. After electromechanical uncoupling with 10 µm 2,3-butanedione monoxime in Tyrode's solution, a 100 µL bolus of the voltage-sensitive dye di-4-ANEPPS (excitation: 482 nm, emission: 686 nm; Biotium, Inc., Hayward, CA, USA) was infused for 5 min in 4.16 mm DMSO to monitor electrical activity via fluorescence changes. The spatial resolution was defined by a subject-specific conversion factor established before the OM experiments and maintained constant throughout the study (pigs 8–12: 0.081, 0.0729, 0.0860, 0.0885, and 0.0906 cm/pixel, respectively). All recordings were acquired at a temporal resolution of 2 ms.

Only one induction event was mapped per LAD pig. To induce VT, a stimulation catheter delivered a train of stimuli consisting of 12 initial pulses at a cycle length of 1000 ms, followed by 5 s of pacing with progressively decreasing cycle lengths from 300 to 150 ms until successful induction was achieved. The stimulation catheter was located in the LV for pig 8, in the left atrium for pigs 9–11, and in the right atrium for pig 12. The OM recordings were processed as previously reported [20,28].

**2.2.2 Computational protocol.** Our implemented AIP was inspired by the work of Arevalo et al. [16]. As shown in Fig 1B, each BiV model was automatically divided into 19 regions using LGE-CMR-based segmentation of the LV and right ventricle (RV). The LV was segmented into 17 regions according to the American Heart Association (AHA) standard and consisted of 6 basal, 6 midventricular, 4 apical segments, and the apex, while the RV was divided into two regions: the apex and the remaining RV (Fig 1B). Based on this partition, the centroid of each segment was calculated, and its nearest endocardial node was identified. Pacing sites were defined as the set of nodes located within a 1 mm-radius sphere centered on each of these endocardial nodes. For each pacing site, an S1-S2 protocol was applied. Four S1 stimuli were applied every 600 ms, and one extrastimulus S2 was applied at 250, 265, 280, or 295 ms after the fourth S1 stimulus (Fig 1E). All stimuli had an amplitude of 80 µA/cm$^2$ and a pulse width of 1 ms.

To maintain computational efficiency while ensuring a representative induction study, a subset of at least 5 well distributed pacing sites (out of the 19 AHA-standardized locations) was selected for each model based on two primary criteria: spatial coverage and proximity to the BZ (Fig 2). Sites were selected to homogeneously span the MI perimeter, ensuring coverage across the superior, inferior, and lateral sides. For each anatomical direction, pacing sites closer to the BZ were prioritized, provided they were located within viable myocardium rather than within non-conductive scar tissue. This strategy ensured that the pacing stimulus was delivered to tissue capable of supporting reentrant activity while capturing the spatial heterogeneity of the proarrhythmic substrate and its response to localized triggers [39,40]. In the LCx and LAD models, the LV apex and RV apex were consistently used as one of the selected pacing sites, respectively.

The resulting electrical activity obtained from stimulating every pacing site was labeled as no reentry (NR), non-sustained VT (nsVT), and sustained VT (sVT). NR indicates cases where no reentrant activity was observed, while nsVT and sVT indicate that reentry vanished earlier than or was maintained for 2 s after the application of the S2 stimulus, respectively. From this categorization, an inducibility score (IS) was defined as $IS = (\sum_{i=1}^{N} c_i)/(3 \times N)$, where $N$ is the number of simulations used for its calculation, and $c_i$ is 0, 1, or 3 if the $i^{th}$ simulation resulted in NR, nsVT, or sVT, respectively. In this way, IS was defined in the range [0,1], with a value of 0 when the $N$ simulations resulted in NR, 0.33 when all simulations resulted in nsVT, and 1 when sVT occurred in all cases.

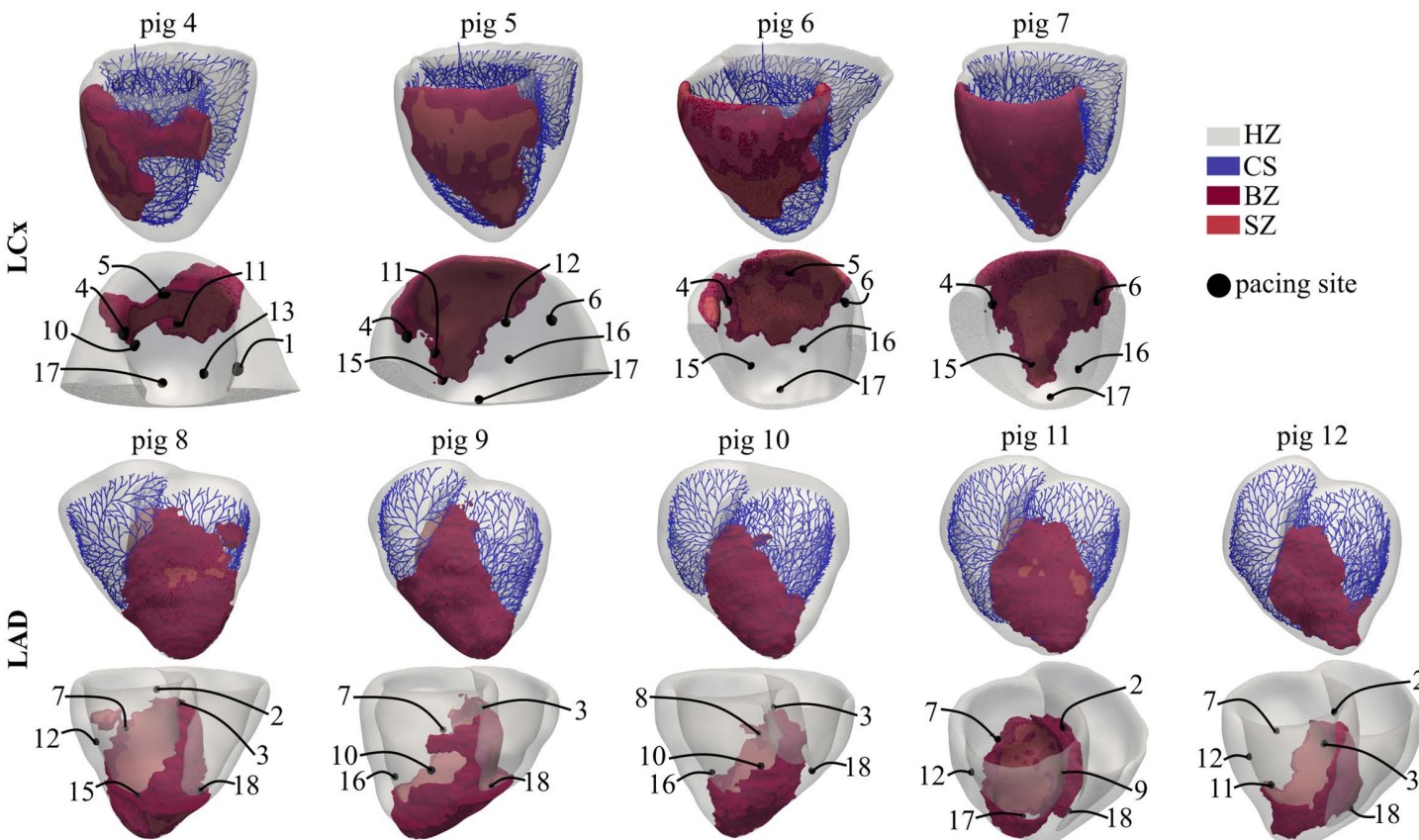

**Fig 2. Porcine-specific BiV models of MI.** BZ, SZ and CS are depicted together with the subset of selected pacing sites used in each pig for application of the AIP. To show the pacing sites in LCx cases, a cut near the LV endocardial septum was performed and a sagittal-like view is presented.

## 2.3 Coupling of BiV and EHT models

A pipeline was implemented to build realistic BiV-EHT models using our previous work as a basis [29]. This pipeline is shown in Fig 3. First, the EHT was represented as a 40x40x1 mm$^3$ squared superficial mesh, and a rigid transformation was used to bring it closer to the MI region of the BiV mesh (Fig 3A). The EHT size was selected based on experimentally achievable dimensions that allow remuscularization of most of the MI region [7,10]. In the high-resolution EHT mesh, contact nodes were defined as those on the face closest to the epicardial surface. These contact nodes were individually projected onto the epicardial surface, with the same displacements applied to the nodes of the non-contacting face. The deformations imposed on the EHT, based on this node-wise contact displacement, resulted in an EHT deformation that followed the epicardial surface of the BiV mesh, as shown in Fig 3B. Subsequently, the EHT was embedded in the BiV model by projecting 0.5-0.6 mm of its thickness into the epicardium (see the right circle in Fig 3B). The projection direction was set as the inverse of the normal direction of the epicardial node closest to the centroid of the EHT mesh. Following embedding, the EHT and BiV meshes were merged using an *XOR* operation in Meshlab [41,42] (Fig 3C). The internal portion of the EHT located within the ventricular wall (blue region in Fig 3C) was removed. To ensure a high-quality volumetric mesh, irregular triangles at the BiV–EHT interface, which would otherwise lead to poor-quality tetrahedralization, were identified based on their aspect ratio and refined using the isotropic explicit remeshing filter in Meshlab [42,43] (Fig 3D).

The BiV-EHT meshes were tetrahedralized using Tetgen [44] (Fig 3E). The resulting tetrahedral meshes presented a bimodal edge length distribution, as shown in Fig 3F for pig 10. The low and high modes of these bimodal edge length distributions were calculated for all BiV-EHT models. A mean value of 187.2 µm was found for the low mode, which corresponded to the mean edge length in the EHT part. In addition, a mean value of 381.9 µm was found for the high mode, which corresponded to the mean edge length in the BiV part. The rationale for this domain tetrahedralization was to ensure simulation accuracy, particularly within the thin EHT and its surrounding regions. Due to its small thickness, the

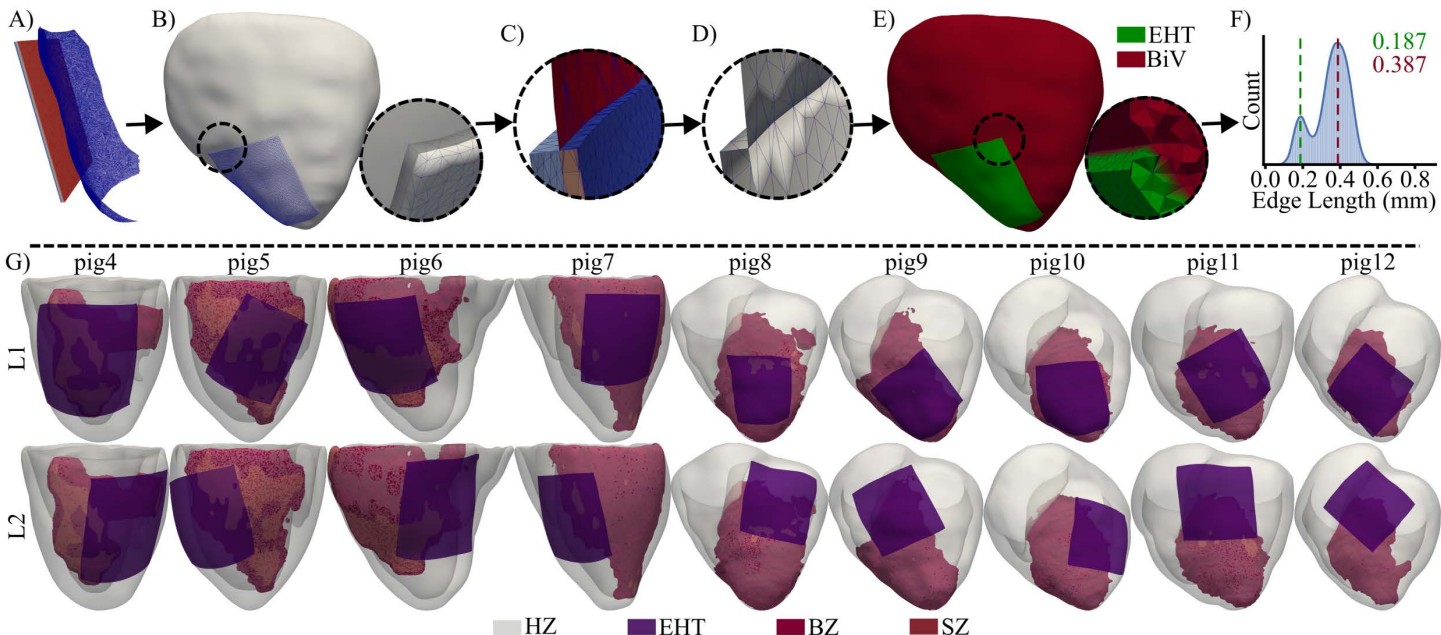

**Fig 3. BiV-EHT modeling.** Pipeline used to generate the BiV-EHT model for pig 10. A- Aligned EHT and epicardial meshes (EHT contact nodes in red). B- Deformed EHT mesh embedded on the BiV mesh (pig 10). C- Integrated BiV-EHT mesh with distinct regions and irregular triangles. Light-blue: outer EHT, blue: inner EHT, red: outer BiV, and pink: inner BiV. D- Cleaned BiV-EHT mesh. E- Tetrahedral mesh with BiV (red) and EHT (green) meshes. F- Bimodal edge length distribution in the BiV-EHT model. G- BiV-EHT models for all pigs and the two EHT locations.

EHT required a higher element resolution, while the overall computational burden was kept to a minimum. The mean numbers of tetrahedrons and nodes for all models were 26 658 338±4 974 043 and 4 512 334±817 799, respectively. These values were reduced to 23 623 451±4 928 526 tetrahedrons and 4 054 310±815 100 nodes after SZ deletion to simulate this region as an insulator (see Sect 2.4). The electrophysiological characteristics and fiber orientations of the BiV part in the coupled BiV-EHT models were interpolated from the corresponding BiV model.

## 2.4 BiV and EHT electrophysiology

The biophysically detailed cellular models of Gaur et al. (pig) [25] and Stewart et al. (human) [45] were used to represent the AP of the porcine BiV models and their corresponding CS [20]. The electrical activity of hiPSC-CMs in the EHT was described by the updated cellular model by Paci et al. [46].

Regional repolarization heterogeneities in HZ were defined by tuning the conductance of the inward rectifier $K^+$ current ($G_{K1}$) in the Gaur et al. [25] cellular model, as this current is the main modulator of the APD in this model [20,25]. In BZ of all models, $G_{K1}$ was reduced to 29% of its default value to match the average prolongation of APD at 90% repolarization ($APD_{90}$) previously observed by us, after chronic MI in pigs [20].

All ventricular regions in the models were considered to have orthotropic conductivity with transverse isotropy. EHT conductivity was set as completely isotropic due to the lack of precise data on connexin-43 distribution in hiPSC-CM cultures; consequently, the EHT was modeled without preferential directionality or a defined fiber architecture. Specifically, the longitudinal diffusion coefficients (LDCs) in HZ, CS, and BZ were set to 0.0013, 0.013, and 0.000882 $cm^2$/ms (equivalent to 0.13, 1.3, and 0.0882 S/m for a membrane surface-to-volume ratio of 1000 $cm^{-1}$ and capacitance per unit area of 1 $\mu F/cm^2$), respectively [20]. The HZ LDC was initially defined based on human studies [47]. However, this value was subsequently calibrated and validated using OM-derived conduction velocities (CVs) from the anterior surface of the porcine heart [20]. Similarly, although the CS LDC was set to achieve CVs characteristic of the human CS (≈ 200 cm/s) [48], it successfully reproduced the experimental, OM-derived maximum activation times and depolarization patterns of the anterior porcine heart under sinus rhythm [20]. The LDC for the CS was reduced near its endpoints using a sigmoid profile to match the ventricular LDC, as proposed by Dux-Santoy et al. [49], thereby enabling retrograde propagation at the PMJs. Bidirectional conduction across the PMJs was achieved (see S1 Fig). A minimal anterograde propagation delay of 1.1 ms was observed, whereas the retrograde delay was zero (see discussion in S1 Section and Sect 4.6). The selected LDC value for BZ corresponded to the average of the individual values required to match *in silico* the CV in the MI region of each of the 5 LAD pigs previously measured by OM [20]. The excitability in BZ was reduced by decreasing the conductance of the fast $Na^+$ current to 38% of its default value. This adjustment follows canine-related experimental observations [50] and aligns with established protocols from previous human and porcine computational studies [16,23,51]. The transverse-to-longitudinal diffusion ratios were set to 0.345 and 0.25 in BZ and HZ, respectively. For BZ, this ratio reflects the reduced anisotropy observed relative to HZ in diffusion-weighted CMRs [20]. The ratio for HZ was based on measurements characterizing transverse anisotropy in healthy myocardium, where CV is greater along the longitudinal axis and lower along the transverse axis [52]. As transmembrane potential ($V_m$) gradients have not been observed in experimental chronic MI electrocardiograms [20,51], the SZ was modeled as an insulator. This was implemented by establishing a zero-flux boundary condition around the SZ by simply excising it, since a Neumann boundary condition is used in the monodomain model used herein (see Sect 2.5). Specifically, all tetrahedra composed exclusively of SZ nodes were removed from the computational domain. For the remaining tetrahedra containing at least one SZ node, those nodes were retagged as the non-SZ nodes within the same tetrahedron. This approach was adopted because removing all elements containing any SZ node would be overly aggressive and could eliminate narrow, low-conducting BZ isthmuses, which are critical substrates for reentrant tachycardias [39,51,53,54]. Overall, modeling the SZ as an insulator avoids the *in silico* generation of $V_m$ gradients between the depolarized HZ and the electrically inactive SZ, which are not experimentally observed in chronic MI [20,51].

All ventricular cell models were prepaced to steady state by applying periodic S1 stimuli (80 µA/cm² magnitude, 1 ms duration, 600 ms period). For the Paci et al. [46] model, prepacing was performed according to the stimulation described in [46] (0.55 nA magnitude, 5 ms duration) with a period of 600 ms. The intrinsic automaticity of the Paci et al. [46] model was preserved without modification. Stimulated prepacing was used to reflect standard preculturing protocols, in which external electrical stimulation is applied to enhance hiPSC-CM maturation [14]. Stabilization of all cellular models was confirmed by verifying that $APD_{90}$ remained constant across consecutive beats. Although repolarization reached a steady state after 800 s, a total of 2000 beats (1200 s simulation time) were simulated to ensure final convergence of $APD_{90}$ to: 310 ms for the CS (under spontaneous automaticity), 196–218 ms across all transmural and apicobasal regions of the HZ, 384 ms for the BZ, and 405 ms for the EHT.

## 2.5 Numerical simulations

First, simulations were conducted to investigate the contribution of MI characteristics and CS to arrhythmic risk (Fig 1F, top panel). In the first group of simulations (G1), the complete AIP was applied to all CS-BiV models. Specifically, S1-S2 stimulation was applied 4 times to each pig model in each selected pacing site, corresponding to the four S2 values of 250, 265, 280, and 295 ms. To qualitatively validate our models, the simulated depolarization patterns calculated for LAD pig models were compared with those obtained from OM experiments for the same pigs when the experimental AIP described in Sect 2.2.1 was applied. In this set of simulations, proarrhythmicity was evaluated as a function of the S2 delay, as well as the location, shape, and size of the MI.

In a second group of simulations (G2), proarrhythmicity was evaluated as a function of the inclusion or absence of CS in the models. Specifically, the LAD and LCx pigs with the highest (HI) and lowest (LI) VT burdens in G1 simulations were resimulated without the CS at an S2 interval of 295 ms. These results were subsequently compared to the reentrant activity observed in the corresponding G1 simulations, which included the CS, at the same S2 interval.

Subsequently, changes in vulnerability to arrhythmias after post-MI remuscularization with EHTs were investigated (Fig 1C and 1F). A third group of simulations (G3) was carried out in which AIP was applied with an S2 of 295 ms before and after the engraftment of EHT in all pigs, and when the EHT was implanted in two different locations. For the central location L1, the EHT was implanted in the center of the MI. In contrast, for the lateral location L2, the EHT was positioned such that half of it overlapped the MI region, while the other half was in contact with HZ (Fig 3G). The rationale for selecting these locations was to balance two competing factors: mechanical support is believed to be maximized when the EHT is placed centrally within the MI; on the other hand, the survival of the EHT could be improved when it is in contact with well-vascularized healthy tissue. In all G3 simulations, the conductivity for immature EHTs was set to 10% (C10) of the value found in the HZ (0.00013 cm²/ms) to match experimental observations. This value was previously validated in studies where simulated maximum activation times and CVs (≈12 cm/s) closely matched OM measurements from *in vitro* EHTs [10,28]. Experimentally measured velocities typically range from 5 to 24 cm/s depending on scaffold architecture and EHT culturing [10,13].

In addition, a fourth group of simulations (G4) was conducted to assess the effects of the electrical maturation of the EHT on proarrhythmicity. Higher maturation was represented by greater cell-to-cell coupling in the EHT. For G4 simulations, a pacing site was selected for each pig model based on the results of the G1 and G3 simulations; subsequently, S1-S2 stimulation was applied. Setting the S2 stimulus at 295 ms for this pacing site, all coupled BiV-EHT models, with EHT located at L1 and L2, were evaluated with the EHT conductivity set to 90% (C90) of the HZ value. This conductivity represents a mature, highly integrated scenario. Computational evidence indicates that matching the host myocardium's conductivity is critical to prevent the EHT from acting as a proarrhythmic substrate [17,26,28–30,55]. Such high functional maturation is increasingly achievable through advanced scaffold designs, including diamond-shaped pores for directional stiffness control or the incorporation of electroconductive materials, which enhance cell alignment, connexin-43 expression, and electrical integration [10,56,57]. Consequently, results from C10 (immature, low conductivity) and C90 (mature,

high conductivity) EHTs were compared for each subject to quantify the effect of post-implantation electrical maturation on overall VT risk.

In all simulations, the monodomain reaction-diffusion equation was numerically integrated using the open-source, in-house solver ELECTRA (v0.6.3) [58]. High-throughput execution was achieved through batch simulation using HER-MES, the HPC Unit of the University of Zaragoza. Computed $V_m$ was saved at a time resolution of 1 ms, while the adaptive integration time step was constrained within the [0.01,0.1] ms range for all *in silico* calculations. In total, 335 simulations were performed, representing a cumulative duration of more than one million milliseconds of electrophysiological activity. Although large-scale simulated time is increasingly common in HPC-based cardiac research, the scale of this study is particularly notable given the high spatial resolution of the BiV meshes and the dense incorporation of electrophysiological heterogeneities. The substantial wall-clock time required and the fine spatial discretization place this work at the forefront of current simulation studies on remuscularization [17,26,59]. Furthermore, by modeling the EHT as directly implanted onto the epicardial surface of the BiV anatomy, we introduce an enhanced level of structural and physiological realism. This detailed representation of the graft-host interface at the organ scale captures a level of complexity not previously addressed in simulations of EHT-mediated remuscularization [17,26], thereby providing a more robust framework for evaluating post-implantation arrhythmogenicity.

## 2.6 Simulated conduction velocities

In Sect 2.4, conductivity values for the CS and HZ were defined based on existing large-mammal literature (canine [52] and human [47]) and validated against our own porcine-specific experimental data (electrocardiogram and OM) obtained under sinus or paced rhythm (see our previous work [20]). Additionally, BZ conductivities were adjusted to reproduce post-MI conduction CVs previously measured via OM under sinus or paced rhythm [20]. In the present work, inclusion of the EHT modified the computational domain, requiring mesh refinement in the remuscularized region (see Sect 2.3). To ensure that the selected conductivities, together with the non-uniform mesh resolution, preserved physiological accuracy, we evaluated the simulated CVs.

CVs were calculated across the epicardial faces of the BiV meshes to maintain consistency with the superficial nature of OM data [20]. The CV magnitude between two nodes was computed as the ratio of their Euclidean distance to the difference in their activation times. For each node, a mean CV vector was calculated within a defined radius [20]. As shown in S2 Fig, multiple radii were evaluated; smaller radii (e.g., 0.6 mm) produced numerical artifacts, specifically infinite CVs (red in S2 Fig), due to the 1 ms activation time resolution and the proximity of nodes with identical activation times, particularly in fast-conducting HZ. Conversely, larger radii yielded more biomimetic CVs in the HZ, with low variation upon further radial increments. Accordingly, a 2 mm radius was selected as the minimum threshold providing consistent, finite CV values in the HZ.

Under this configuration, CVs were computed for the final S1 stimulus in pig 6 (pacing site 15) across all modeling scenarios: the baseline MI model without EHT, and models with the EHT positioned at L1 and L2 under C10 and C90 conductivity settings (Fig 4A, 4B, and 4C). In the HZ, simulated CVs ranged from 69–79 cm/s. Despite differences in mesh resolution among cases, HZ values remained consistent and, notably, closely matched the experimental healthy porcine CV of 74.2 cm/s, previously measured by OM under sinus and paced rhythm [20]. Lower CV values were primarily attributable to the presence of the slower-conducting EHT (notably the L2-C10 configuration) rather than to variations in mesh edge length. Indeed, as shown in Fig 4A CVs within the finely meshed EHT region were comparable to those of the underlying myocardium (BZ or HZ) and were consistent with the values obtained in more distant, coarser-meshed equivalent (BZ or HZ) regions. Similarly, CVs in the BZ were robust to changes in mesh resolution. Values were nearly identical (27 cm/s for C10; 34 cm/s for baseline MI and C90) regardless of whether the EHT was implanted directly above (L1), lateral to (L2), or absent from the region. These simulated CV values agree with the experimental porcine range of 27.9–43.6 cm/s measured in the MI region by OM under sinus and paced rhythms [20].

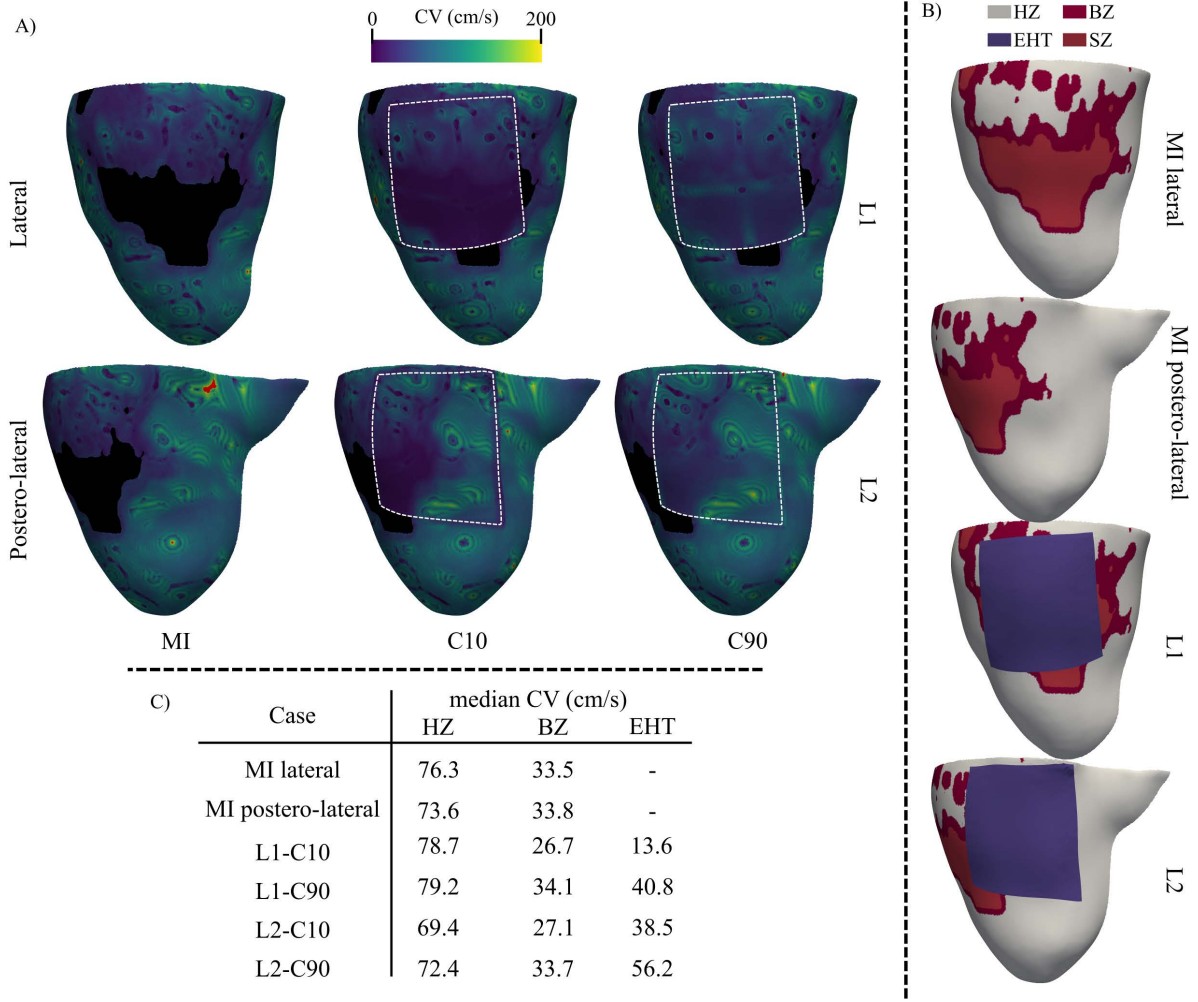

**Fig 4. Simulated CV with and without the EHT.** A- Node-wise CV maps for pig 6 following the last S1 stimulus applied at pacing site 15 without the EHT (MI case) and when the EHT was implanted in different locations (L1-L2) and assigned different conductivities (C10-C90). Undefined values due to infinite CV (matching activation times among nodes) are depicted in red. B- Segmentation of the analyzed ventricular faces, highlighting HZ, BZ, and EHT regions. C- Median CVs obtained for different regions with and without the EHT.

The CV in the EHT was influenced by both its intrinsic conductivity and the underlying myocardium, suggesting that the host tissue effectively drove EHT depolarization, as reported in [17,29,30]. At the L1 position with low conductivity (C10), the EHT exhibited a CV of 13.6 cm/s, consistent with reported *in vitro* EHT values (5–24 cm/s) [10,13]. Because the EHT at this location predominantly overlies non-conducting SZ or slow-conducting BZ, the simulated CVs naturally approximated *in vitro* values. In contrast, when assigned adult-like conductivity (C90), the CV in the EHT increased, particularly at the L2 position where it overlies fast-conducting HZ, approaching healthy myocardial velocities.

Overall, this analysis confirms that the simulated CVs accurately reproduce porcine-specific experimental values under healthy and MI conditions, in both sinus and paced rhythms, as well as *in vitro* EHT dynamics, despite the non-uniform, bimodal mesh resolution. These results provide a solid foundation for our arrhythmia inducibility assessment. This quantitative accuracy is further supported by qualitative validation, in which simulated arrhythmic patterns were compared with subject-specific, OM-derived arrhythmic events (G1 simulations). Importantly, no additional adjustments to conductivity

parameters were made using these OM data under arrhythmia, ensuring a clear separation between model calibration (based on OM of sinus and paced rhythms) and evaluation (based on OM of reentrant activity).

## 3 Results

### 3.1 Experimental and simulated arrhythmic patterns in MI

The experimental arrhythmic patterns mapped by OM in LAD pigs were qualitatively compared with the results from G1 simulations using the CS-BiV models of pigs 8–12. Results for pigs 8, 10, and 11 are shown in Fig 5, where the presence of the CS is visible upon magnifying the simulation images. In pigs 8 and 11, a counterclockwise reentry was observed in the anterior view of the OM recordings. These experimental patterns were reproduced by the *in silico* simulations when AIP was applied with an S2 of 295 ms at pacing site 12 for pig 8 and at pacing site 7 for pig 11. Generally, the depolarization wave traveled in the LV from the apex to the base, crossed to the basal RV, and propagated downward to the apical

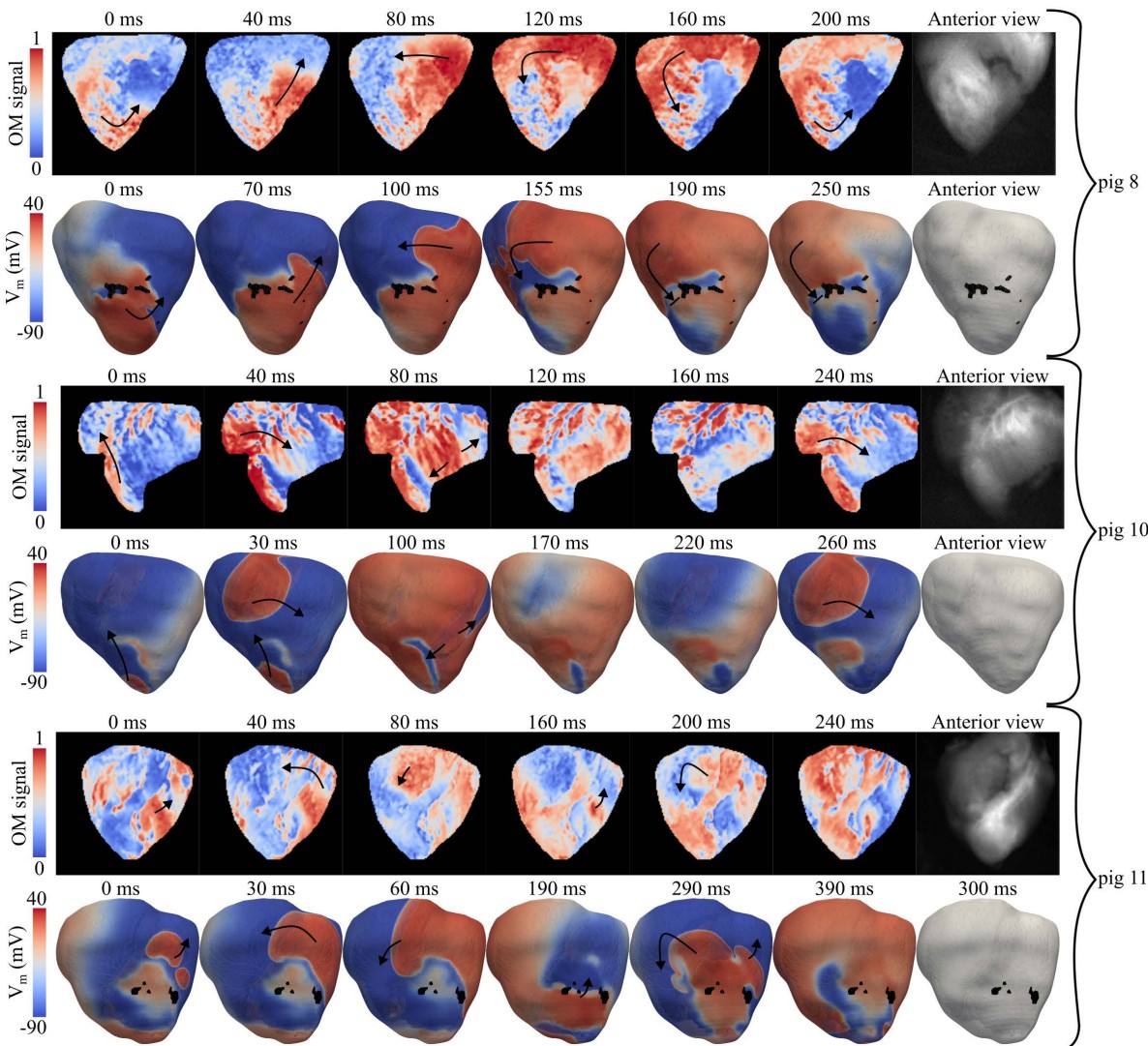

**Fig 5. Reentrant activity in LAD pigs.** VTs generated after application of the experimental (top) and *in silico* (bottom) AIPs.

RV. In pig 10, the experimentally observed clockwise reentry was replicated *in silico* by stimulating pacing site 3 with an S2 of 280 ms. For this pig, the epicardial activation of the anterior view progressed sequentially from the LV apex to the RV base, then to the medial LV, and subsequently, the depolarization reached the laterobasal LV and the LV apex. Simulations also reproduced the counterclockwise reentrant activity observed in LAD pigs 9 and 12, with an S2 of 295 ms applied at pacing sites 16 and 12, respectively (see S1 Video).

While experimental VTs were sustained in all infarcted pigs, the corresponding *in silico* simulations exhibited a mix of dynamics: nsVTs were observed in models for pigs 8–10, whereas sVTs were successfully reproduced in pigs 11–12. As an example, in pig 8, a single reentry was observed because the depolarization wave was stopped when traveling from the basal RV to the apical LV (see the simulated reentry of pig 8 in Fig 5). In spite of minor differences, experimental and simulated reentrant dynamics showed good qualitative agreement in all cases. Experimental and simulated arrhythmic patterns are provided for all pigs in S1 Video.

### 3.2 Quantification of VT inducibility after MI

Fig 6 summarizes the results of the G1 simulations, while the full set of results is provided in S1 Table. When AIP was applied and the stimulation captured, considering all pacing sites and all S2 values, pigs 8, 9, 10, 11, and 12 achieved IS values of 0.30, 0.37, 0.54, 0.7, and 0.1, respectively. Specifically, a single sVT was generated for pigs 8, 9, and 12, while 4 and 5 sVTs were found for pigs 10 and 11, respectively (Fig 6, left panel). Of all pigs with simulated LAD occlusion, pig 11 was the one with HI, while pig 12 was the one with LI. In particular, for pig 12, the application of AIP in 84.6% of the paced sites failed to develop a reentry. In addition, LCx pigs were found to be less vulnerable to arrhythmias (Fig 6, middle panel). When capturing, only nsVTs were generated in pigs 5 and 7 (IS: 0.33), and NRs were observed in all cases in pigs 4 and 6 (IS: 0) (Fig 6, left panel).

Regarding the S2 stimulus, the stimulated tissue was still refractory when the extrastimulus was applied 250 ms after the last S1 stimulus, preventing capture (Fig 6, middle panel). As can be observed in the middle and right panel of Fig 6, inducibility increased when the S2 delay increased, with contained increments in IS observed in LCx pigs. IS values

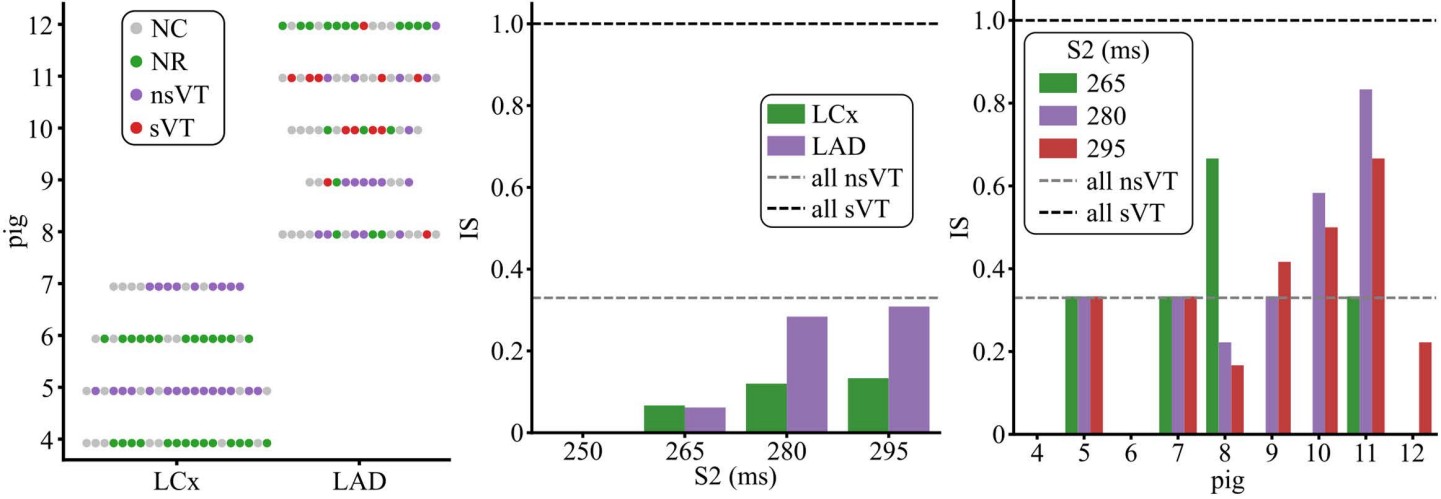

**Fig 6. Post-MI inducibility for all nine CS-BiV models when all pacing sites were tested (G1 simulations).** The left panel shows the pig-wise VT inducibility results as a function of MI location when S2 values of 265, 280, and 295 ms were used. The middle panel depicts the IS for both MI types as a function of all S2 values. The right panel presents the pig-wise IS for all S2 values. As detailed in Sect 2.2.2, the simulations where the stimulation did not capture (NC) were not considered into the IS computation.

of 0.06, 0.12, and 0.13 were measured for S2 values of 265, 280, and 295 ms, respectively, in LCx cases. In LAD pigs, IS increased sharply from 0.062 to 0.28 when S2 changed from 265 to 280 ms, and slightly from 0.28 to 0.31 when S2 changed from 280 to 295 ms, respectively. Thus, the S2 delay had a greater effect on inducibility in LAD pigs than in LCx pigs. Only in pig 8 was inducibility reduced as the S2 delay increased from 265 to 280 ms (Fig 6, right panel). For both types of MI, an S2 of 295 ms led to the highest inducibility. Therefore, other groups of simulations were conducted using this delay.

### 3.3 CS effect on arrhythmogenesis after MI

In the G2 simulations, LCx and LAD pigs with low and high inducibility in G1 were considered to evaluate the effect of CS on arrhythmogenesis. Some simulation examples are provided in S2 Video and the complete set of results for the G2 simulations are provided in S2 Table.

For the group of LCx pigs:

- *Pig 6:* Exemplified a case of low inducibility in which CS played a fundamental role in blocking the reentry. As shown in the first row of Fig 7, application of AIP at pacing site 6 generated an S2-derived depolarization wave that initially propagated from the anterior to the posterior LV via the apex. When the CS was present, a CS-derived epicardial breakthrough intercepted this S2-derived activation in the laterobasal region, effectively preventing reentry. In contrast, without the CS, the same S2-derived activation encountered no such breakthrough, allowing sVT to develop. This outcome is further corroborated by the $V_m$ traces recorded after the final S1 stimulus (see right column of Fig 7). With CS, the laterobasal (anterior: purple AP; posterior: yellow AP) and apical (green AP) regions were activated almost simultaneously by the last S1 stimulus compared with the condition without CS. In this scenario, the anterior laterobasal region (purple) remained refractory upon the arrival of the S2-derived wavefront, resulting in a unidirectional block. Without CS, this led to counter-clockwise reentry, as evidenced by the sequential depolarization observed in the green, yellow, and purple $V_m$ traces. With CS, however, an epicardial breakthrough caused simultaneous depolarization of the anterior (purple) and posterior (yellow) laterobasal regions, thereby blocking reentry.

- *Pig 7:* This pig was selected as an example of high inducibility (pig 5 was not selected because nsVT in pig 5 propagated through the low-conducting laterobasal isthmus in the LV and might disappear in a whole-heart simulation, as described in Sect 3.6). The application of AIP at pacing site 17 of pig 7 resulted in nsVT in the simulation with CS and in NR in the simulation without CS. In the former case, the CS enabled the MI to function as a capacitor: the faster depolarization and subsequent repolarization of the MI region permitted gradual loading of this region with S2-derived activation, followed by the release of the capacitive energy stored in the MI region from the apex back to the HZ once the refractory-based impedance in the HZ was low enough to allow the MI to be discharged, thereby producing reentry. Without CS, the S2-derived activation did not penetrate the MI region, which remained refractory. As shown by the corresponding $V_m$ traces, the CS promoted more synchronous S1-induced activation of the MI (purple AP) and apical (green AP) regions. This synchronization allowed the S2-induced depolarization to reach a less refractory MI region (see $V_m$ traces at 390 and 490 ms). Consequently, the MI became activated and sustained depolarization, as evidenced by the S2-derived apical-MI-apical activation sequence observed in the $V_m$ trace with CS, which was absent without CS.

Similarly to the LCx cases, the presence or absence of CS determined an inversion in reentry occurrence in LAD pigs:

- *Pig 11:* Associated with HI among LAD pigs, the CS initiated delayed endocardial activations when pacing was applied at site 7. These activations propagated transmurally, producing epicardial breakthroughs at 590 and 840 ms after the last S1 stimulus, thereby sustaining VT (see the $V_m$ maps and local traces in the third row of Fig 7). These epicardial breakthroughs are further supported by the higher activation rate observed in the anterobasal region (purple AP, third row of Fig 7) compared with the apical region (green AP, third row of Fig 7). Beyond its

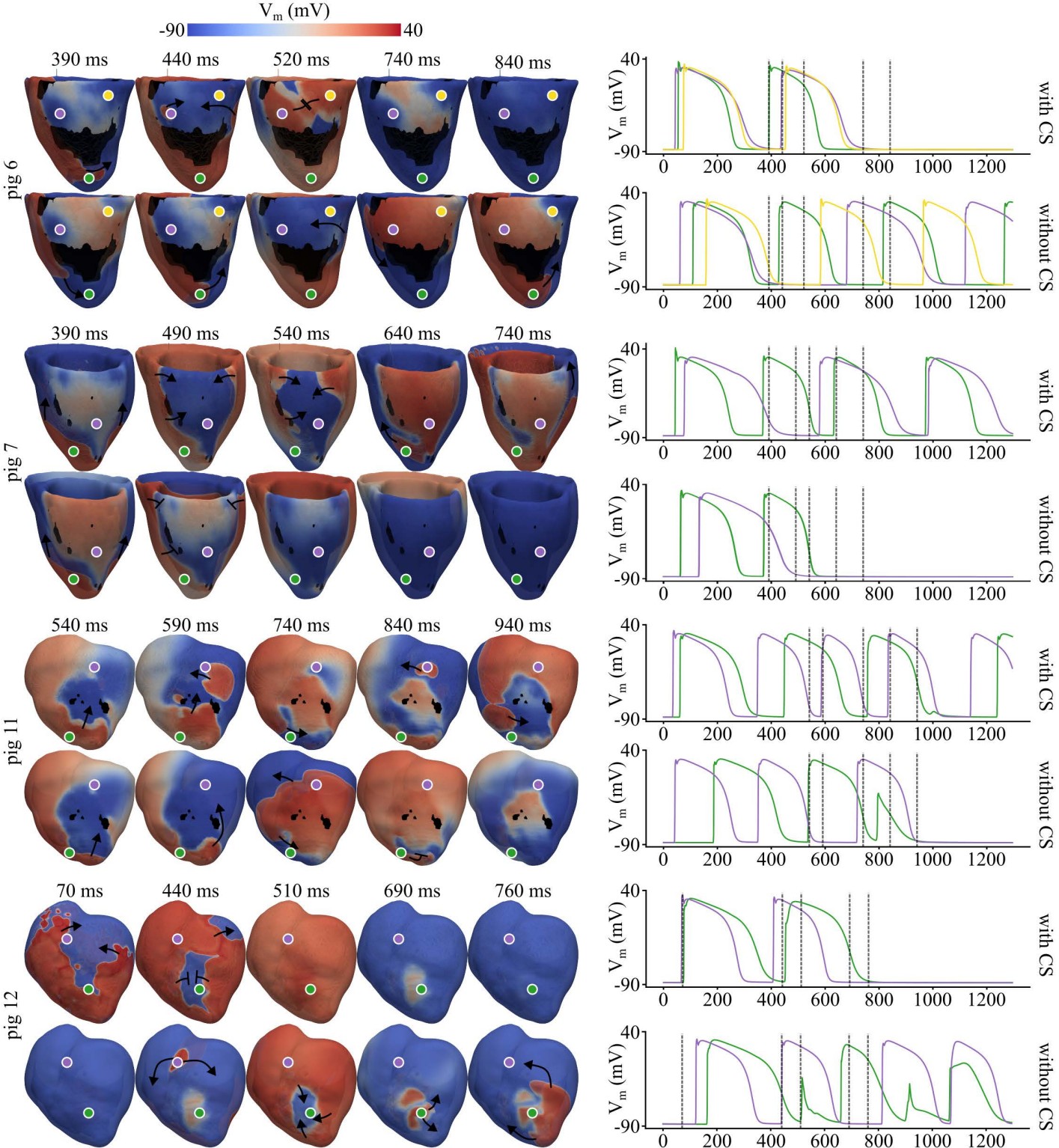

**Fig 7. Changes in proarrhythmicity associated with CS (G2 simulations).** $V_m$ maps and corresponding local traces are shown in the left and right columns, respectively. Results are presented for pig 6 (LCx LI) stimulated at pacing site 6, pig 7 (LCx HI) at pacing site 17, pig 12 (LAD LI) at pacing site 3, and pig 11 (LAD HI) at pacing site 7, displayed in the first through fourth rows, respectively. Time is referenced to the application of the last S1

stimulus; accordingly, the premature S2 stimulus was delivered at 295 ms in all cases shown. $V_m$ traces were recorded at the following node IDs: pig 6 - 1923820 (green), 259964 (purple), and 1249245 (yellow); pig 7 - 564196 (green) and 1528977 (purple); pig 11 - 1638582 (green) and 3590756 (purple); and pig 12 - 2597575 (green) and 3205945 (purple). The vertical, dashed black lines in the $V_m$ traces correspond to the time points of the $V_m$ maps.

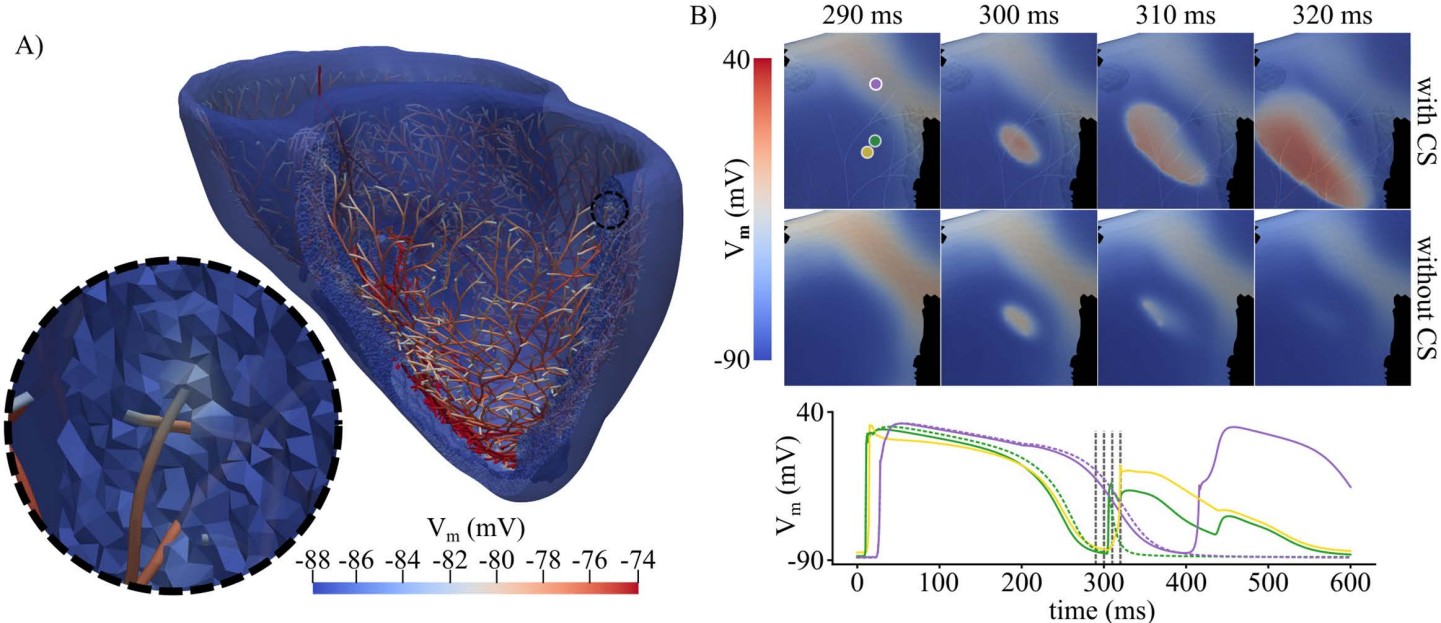

**Fig 8. Effect of CS intrinsic cellular properties.** A- Diastolic potential difference between CS and BiV domain of pig 11, before applying the fourth S1 stimulus at pacing site 7. B- S2-stimulus propagation when applied at pacing site 4 in pig 6 with and without the CS. $V_m$ maps (top) and local traces (bottom) are depicted. $V_m$ traces were obtained at myocardial nodes in the HZ (ID 1576842, green waves) and BZ (ID 576376, purple waves) when the CS was present (solid lines) or absent (dashed lines), and for the PMJ (CS endpoint ID 5873, yellow wave) near the activation. The vertical, dashed black lines correspond to the time points of the $V_m$ maps at the top.

function as an *electrical bypass*, the CS also influenced VT dynamics through other mechanisms linked to its intrinsic cellular properties. A weak, initially undeveloped depolarization wavefront was reenergized in regions adjacent to the PMJs and the MI (see Fig 8B, which illustrates this phenomenon for the S2 stimulus applied at pacing site 4 in pig 6). During diastole, the CS exhibited a slightly higher resting potential than the surrounding myocardium, elevating $V_m$ in nearby ventricular PMJ nodes (Fig 8A). Moreover, because the CS had a shorter APD than the BZ (Sect 2.4), myocardial tissue surrounding the MI displayed reduced APD compared with the condition without CS. As illustrated in Fig 8C, $V_m$ traces from the HZ near the MI and BZ showed shorter APDs in the presence of CS. Additionally, stimulation of the PMJs, even when originating from weakly depolarizing myocardial nodes, can suffice to trigger PMJ activation, given the higher excitability of the CS relative to the myocardium. This, in turn, reinforces the initially weak depolarization of the surrounding myocardial tissue. Collectively, these mechanisms suggest a facilitated propagation of depolarization in the presence of CS, thereby promoting VT sustainability. In contrast, without CS, reenergization of weak depolarization wavefronts and, more importantly, *electrical bypass*-mediated epicardial breakthroughs were absent (see the third row of Fig 7), and VT became non-sustained, terminating approximately 1 s after the last S1 stimulus. As illustrated in the corresponding $V_m$ traces, depolarization of the apical region was inhibited because the tissue remained refractory, as evidenced by the final undeveloped AP in the green $V_m$ trace.

- *Pig 12:* Exhibiting the lowest inducibility among LAD pigs, the presence of CS created an *electrical bypass* between the septal base and the apex of both ventricles when AIP was applied at the septal base (pacing site 3). The $V_m$ traces of the septal base (purple AP) and apical region (green AP) began almost simultaneously in the presence of CS. This bypass enabled earlier depolarization of the MI region compared with the condition without CS (see the bottom row of Fig 7). Consequently, the S2 stimulus arrived at a fully repolarized MI 440 ms after the last S1 stimulus, resulting in NR. Without CS, a *capacitive effect* similar to that described for pig 7 with CS was observed. The S2-derived depolarization wave partially stalled within the MI and gradually penetrated this region. From the $V_m$ traces and the supporting S2 Video, the complex reentrant activity in this case can be appreciated. The apical MI region (green AP) initially exhibited incomplete, transient depolarizations before eventually undergoing full depolarization, as reflected by the transition from undeveloped to fully developed $V_m$ upstrokes. Reentry subsequently occurred when the energy temporarily stored in the apical MI was released back into the HZ once excitability was restored. This initial reentrant event triggered a complex, polymorphic reentrant pattern that continued to sustain VT (see S2 Video).

In general, the presence or absence of CS markedly influenced arrhythmogenesis. Notably, in two cases the S2 stimulus failed to capture in the absence of CS (see VT inducibility in the left panel of Fig 9). Mechanistically, this loss of capture was attributable to the absence of PMJ-mediated reenergization of a weak depolarizing wavefront, as previously described (Fig 8B). As shown in the right panel of Fig 9, removal of CS decreased IS in the HI cases for both MI types (from 0.33 to 0.11 in LCx and from 0.66 to 0.33 in LAD pigs), but increased IS in LI cases (from 0 to 0.25 in LCx and from 0.22 to 0.44 in LAD pigs). These findings suggest that the electrical bypasses provided by the CS can generate breakthroughs or modulate activation of slow-conducting, long-APD regions, leading to subsequent reactivation of the surrounding non-refractory tissue and thereby either sustaining or impeding VT. Consequently, although case-based, these findings highlight the importance of accurately modeling the CS to ensure a robust and reliable evaluation of ventricular arrhythmogenesis.

### 3.4  Impact of EHT implantation location on arrhythmogenesis after MI

Fig 10 shows the effect of EHT engraftment location on arrhythmogenesis. IS values after MI and subsequent EHT implantation highlight a strong dependence of arrhythmogenesis on the occluded artery (see Fig 10A). In LCx pigs, EHT at L1 markedly increased IS from 0.16 to 0.49, while implantation at L2 resulted in a smaller rise (IS = 0.35). In LAD pigs,

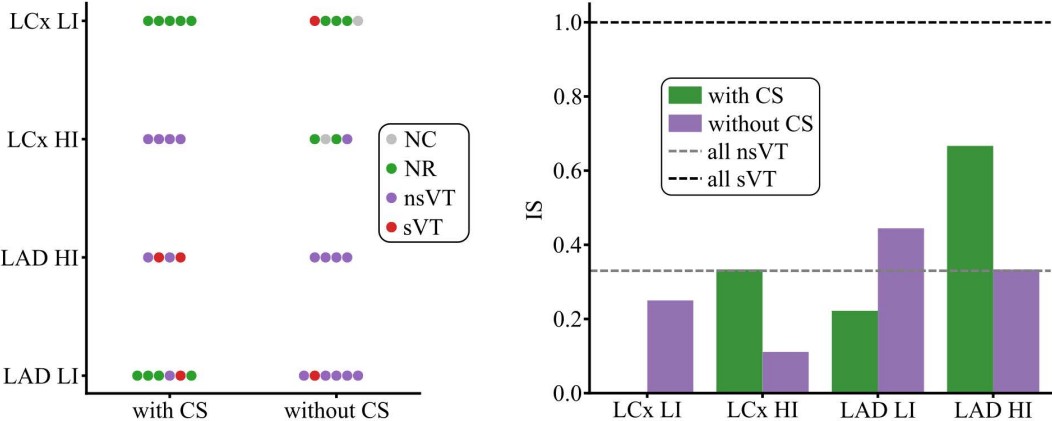

**Fig 9. Effect of CS presence or absence on inducibility (G2 simulations).** VT inducibility (left) and IS (right) computed with and without CS for LI and HI pigs of each MI type (LAD, LCx). LCx LI: pig 6, LCx HI: pig 7, LAD LI: pig 12 and LAD HI: pig 11.

PLOS Computational Biology

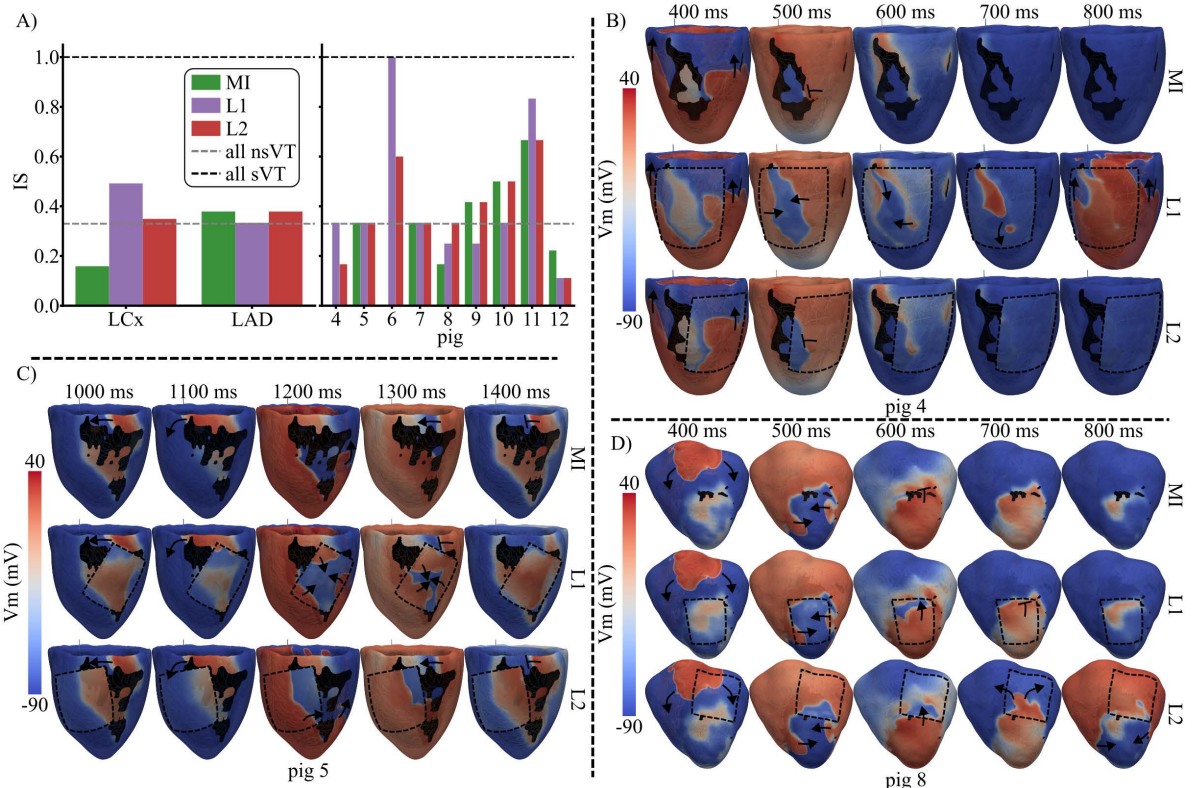

**Fig 10. Effect of EHT engraftment location on arrhythmogenesis (G3 simulations).** In A, IS is presented for BiV MI models and for BiV-EHT MI models, with EHT located at L1 and L2. The left panel displays global results for LAD and LCx pigs, while the right panel shows results for each pig. In B, C and D, $V_m$ maps are presented for pigs 4, 5, and 8, with the EHT outlined (dashed black line). The depolarization-repolarization dynamics after AIP application are presented for MI (top), MI with EHT in L1 (middle) and MI with EHT in L2 (bottom). The pacing sites are 17 for pig 4, 6 for pig 5 and 3 for pig 8. The timings of the $V_m$ maps refer to the time of application of the last S1 stimulus.

IS was slightly lower after implantation at L1 (IS = 0.33) compared with both pre-implantation and implantation at L2 (IS = 0.38), as can be seen in Fig 10A. S3 Table collects all the results for the G3 simulations.

Analyzing individual LCx pigs, IS remained stable in pigs 5 and 7 after MI and following EHT implantation at both L1 and L2. In pigs 4 and 6, EHT remuscularization increased IS, and its highest value was always found in the L1 position. For pigs 4, 5, and 6, the EHT extensively covered the SZ in the L1 position, which generated new conductive pathways between the basal and apical regions of the LV free epicardial wall. This phenomenon is illustrated for pig 4 in Fig 10B, for pig 5 in Fig 10C, and for pig 6 in Fig 11A.

Specifically, in pig 4, the S2 stimulus initiated at pacing site 17 propagated first in the apico-basal direction and then reached the central island of BZ, generating reentry when the EHT was located at L1 (Fig 10B, middle row). In contrast, NR was observed for the same pacing site both after MI and with EHT at L2, as the S2-derived depolarizing wave at the latero-basal LV was unable to cross to the central BZ due to the lack of EHT-mediated electrical bridging. A similar phenomenon occurred in pig 6. When AIP was applied at pacing site 15, CS blocked reentry after MI, as shown in Sect 3.3. A new reentrant pathway was generated in the base-to-apex direction when EHT was implanted at locations L1 and L2, as can be observed for L1 in the upper row of Fig 11A. This pig exhibited the largest IS increase after EHT engraftment. Across all five pacing sites assigned to this pig, AIP simulations yielded no reentry before remuscularization (MI: 5 NR) but resulted in reentry at all sites after remuscularization (EHT-L1: 5 sVTs, EHT-L2: 3 nsVTs and 2 sVTs). Electrical bridging

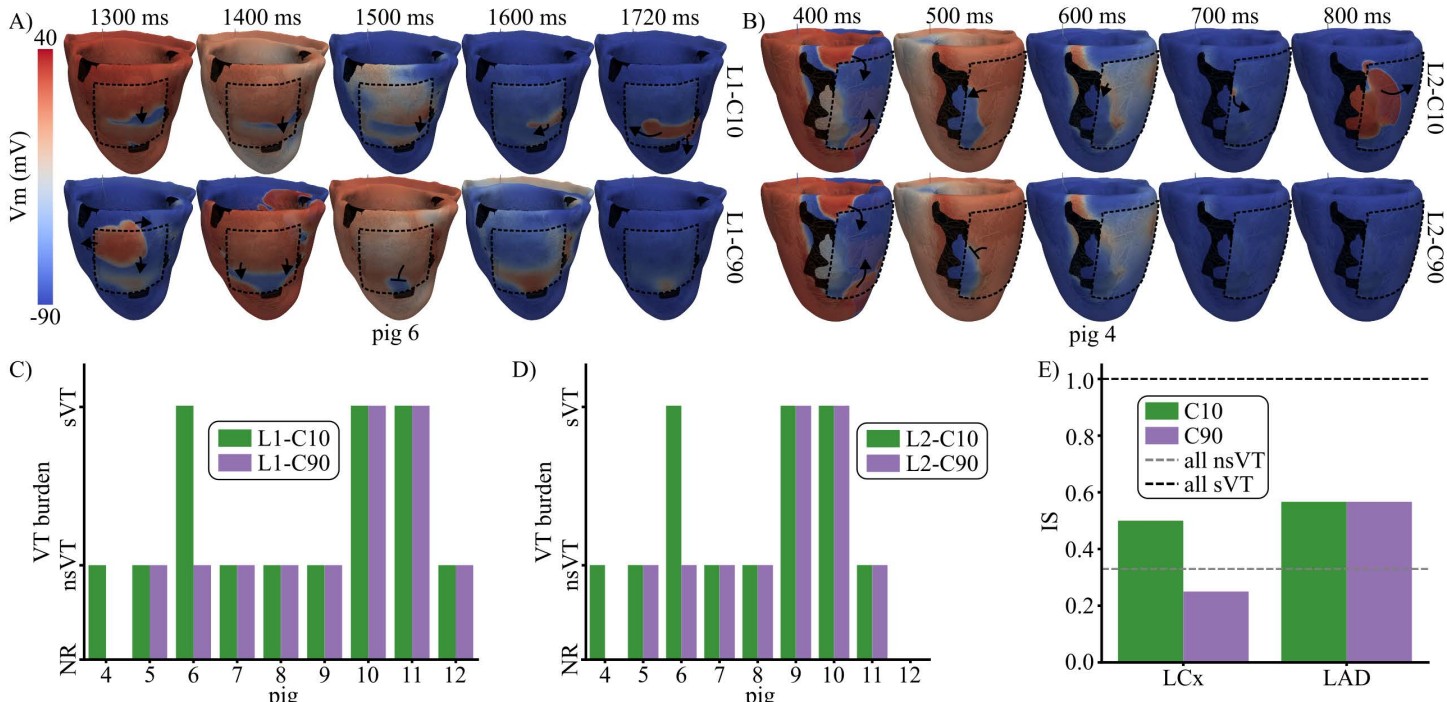

**Fig 11. Effect of EHT electrical maturation on arrhythmogenesis (G4 simulations).** AIP-derived electrical activity is shown for pig 6 (A) and pig 4 (B) when low (top) and high (bottom) EHT (dashed black line) conductivities were used. The pacing sites were 15 for pig 6 and 1 for pig 4. The timings refer to the time of application of the last S1 stimulus. The VT burden was obtained per pig using different EHT conductivities and locations L1 (C) and L2 (D) for a single pacing site. Panel E shows IS for conductivities C10 and C90 of the EHT in MI groups LCx and LAD.

by the EHT was also observed in pig 5 at the L1 location (see Fig 10C), although IS remained unchanged regardless of EHT presence or location (Fig 10A). This stable IS was probably dependent on the fact that this pig persistently presented a reentry blocking the depolarizing wave traveling from the base to the apex through the electrical bridge associated with the EHT, as illustrated in the middle row of Fig 10C.

Unlike LCx pigs, LAD pigs exhibited high variability in proarrhythmic responses following EHT implantation and EHT location (see Fig 10A). Interestingly, in LAD pigs, the implantation of the EHT in L1 reduced IS in 3 of the 5 pigs (pigs 9, 10, and 12) and increased it in the other two pigs. When the EHT was implanted in L2, IS was maintained in pigs 9–11, reduced in pig 12, and increased in pig 8. As an example, the depolarization-repolarization dynamics computed when AIP was applied at pacing site 3 in pig 8 are presented in Fig 10D. Here, similar electrical activity was observed after MI and after EHT implantation in L1, with S2-derived excitation penetrating the MI region and blocking in the mid-epicardial LV. In contrast, implantation in L2 led to reentry, as S2-derived excitation propagated into the highly excitable EHT and continued through the weakly depolarized wavefront, which had previously acted as a conduction barrier.

## 3.5  Contribution of EHT electrical coupling to arrhythmogenesis after MI

In the G4 simulations, EHT electrical maturation was modeled by a variation in its conductivity. As shown in Fig 11, increasing the EHT conductivity from C10 to C90 either decreased the VT burden or kept it at baseline level.

Fig 11A illustrates this effect for two different pigs and EHT locations (pig 6, location L1; pig 4, location L2). In the case of pig 6 and location L1, increasing the conductivity of the EHT to C90 led to a homogeneous fast depolarization and subsequent repolarization of the EHT area above SZ, which prevented an EHT-mediated slow depolarization wave from

maintaining the VT observed for the conductivity C10 of the EHT (Fig 11A, 1720 ms after the last S1 stimulus). A similar phenomenon was observed in the case of location L2 for the same pig 6, where the faster activation of the EHT transformed sVT into nsVT (see S4 Table).

For the case of pig 4 and location L2, increasing the conductivity of the EHT to C90 led to a smoother depolarization wavefront (Fig 11B, 600 ms after the last S1 stimulus). For the conductivity C10 of the EHT, a small self-standing activation was detached from the depolarizing wave and arrived at the peninsular BZ area, which maintained the VT, as a new depolarization wavefront was generated and amplified from this region (top Fig 11B). Increasing the conductivity of the EHT to C90 prevented any reentry from occurring in both EHT locations.

For the remaining LCx pigs (5 and 7), as well as for all LAD pigs, increasing EHT conductivity had no effect on the VT burden compared to the low EHT conductivity setup, as can be observed in Fig 11C and 11D.

When grouped by infarct type (LAD and LCx), higher EHT conductivity reduced the vulnerability to arrhythmias for LCx infarcts (IS from 0.5 for C10 to 0.25 for C90) but had no effect in LAD infarcts (IS of 0.57 for C10 and C90) (Fig 11E). S4 Video presents the complete electrical activity for pigs 4 and 6 after implanting the EHT in locations L2 and L1, respectively.

### 3.6  Role of MI characteristics on arrhythmogenesis before and after EHT implantation

The overall MI size, encompassing both SZ and BZ, was generally larger for LAD pigs than for LCx pigs, as can be observed in S5 Table. The ratio between MI and BiV volumes was 8.8%, 8.0%, 10.6%, and 14.0% for LCx pigs 4–7, and 30.2%, 27.1%, 16.6%, 20.8%, and 13.4% for LAD pigs 8–12, respectively.

For LAD pigs 8–11, a positive correlation between the volume of MI occupied by BZ and the value of IS (considering all pacing sites and all S2 values when the stimulation captured) was observed. Both IS (see Sect 3.2) and the BZ/MI ratio increased progressively from pig 8 to pig 11 (BZ/MI: from 20.7% for pig 8 to 46.5% for pig 11). Pig 12, however, had the lowest IS value of all LAD pigs, despite having an intermediate BZ/MI volume ratio (36.2%). For LCx pigs, pig 7 showed both the highest IS value and the highest BZ/MI volume ratio (45.5%). Pig 5 had the same IS value as pig 7 (the highest among LCx pigs); however, it presented a low BZ/MI volume ratio of 20.8%. This could be explained by the fact that the reentry observed in pig 5 spread along a slow-conducting isthmus located in the latero-basal region of the LV (Fig 10C), suggesting that it could disappear in a simulation of the entire heart in which the addition of the outflow tracts and the atrial tissue alters the isthmus.

In the context of MI remuscularization with EHT, we observed that the size of the epicardial SZ correlated with the magnitude of IS induced by EHT implantation. For LCx pigs, the epicardial SZ was higher in pigs 4–6 than in the other pigs (see Figs 2, 3G, and S5 Table). The ratio between the epicardial area of the SZ and the entire MI was 48.6%, 42.9%, 47.4%, and 3.8% for LCx pigs 4–7, respectively. Correspondingly, pigs 4 and 6 experienced the greatest increases in arrhythmia vulnerability after EHT implantation, which were reduced by placing the EHT in location L2 and increasing the conductivity of the EHT in either of the two locations. Pig 5, despite a high epicardial SZ/MI ratio, showed no IS changes. In this case, the EHT created a new reentrant pathway from the base to the apex. However, unlike pigs 4 and 6, this EHT-derived reentry was blocked by the reentry that occurred before the engraftment of the EHT (Fig 10C). In LAD pigs, epicardial SZ/MI ratios were generally low, ranging from 0% (pigs 10 and 12) to 15.1% for pig 8.

## 4  Discussion

### 4.1  *In silico* models qualitatively reproduce experimental porcine-specific arrhythmic dynamics

Our *in silico* simulations qualitatively reproduced the experimental dynamics observed in OM recordings of Langendorff-perfused LAD porcine hearts, which consistently exhibited VT induction, predominantly sVT. This agreement supports the robustness and reliability of the proposed modeling pipeline. In simulations, nsVTs were observed in pigs 8–10, whereas

sVTs were reproduced in pigs 11–12, consistent with experimental observations in which sVT was always induced. We attribute the absence of sustained VT in pig models 8–10 to the more intensive experimental induction protocol, which used a greater number of pulses and a wider range of consecutive S2 trains than the simulations, although it was delivered at fewer anatomical locations.

The reduced reentry speed observed in simulations compared with experimental rotors (Fig 5 and S1 Video) likely reflects the use of averaged conductivity across the HZ and BZ and across models. This spatial homogenization may have reduced local CVs relative to the heterogeneous structure of the experimental tissue. To define BZ conductivity, which was generalized to all LAD and LCx models, we averaged the individual conductivity values required to computationally reproduce the CV observed in the MI region of the five LAD pigs, as previously characterized by OM under sinus and paced rhythm [20]. Subject-specific parameterization was not feasible for LCx pigs due to a lack of experimental data [20]. Similarly, CS distributions were assigned based on spatial benchmarks but lacked subject-specific experimental validation [20].

Despite these limitations, the G1 simulations qualitatively captured the experimental arrhythmic dynamics without individualized adjustment of electrophysiological parameters or induction protocols, supporting the robustness of the modeling framework. Future studies will assess whether replicating the exact experimental pacing sequence *in silico* and performing fine-grained, subject-specific electrophysiological calibration (e.g., using OM under VT) enhance the stability of reentrant circuits in models 8–10 and better reproduce the reentry speed observed in all LAD models.

## 4.2 An *in silico* protocol for post-MI arrhythmia vulnerability testing

Arrhythmia inducibility was assessed using an *in silico* AIP inspired by previous approaches, such as the Virtual Heart Arrhythmia Risk Predictor [16], but streamlined by employing fewer pacing sites, a shorter S1 train, and a single S2 stimulus delivered at four coupling intervals to balance physiological fidelity and computational tractability. Specifically, Arevalo et al. [16] applied eight S1 stimuli followed by three trains (S2-S3-S4) of premature extra-stimuli. In their study, each premature train was initially delivered 300 ms after the last S1 stimulus and shortened in 10 ms steps until loss of capture or arrhythmia induction. In contrast, we used four S1 stimuli followed by a single premature S2 stimulus (at 250, 265, 280, or 295 ms). The motivation for this reduction was that, under deterministic modeling, electrophysiological activity stabilizes after only a few beats, thereby justifying a shorter pacing sequence. In line with this, Azzolin et al. [60] showed that as few as two stimuli can induce arrhythmia when delivered at the effective refractory period of the pacing site, and that additional premature stimuli rarely increased arrhythmia inducibility in computational atrial models. Accordingly, our protocol prioritizes identification of inducible substrates over exhaustive characterization of the full vulnerability window.

Furthermore, we adopted a more targeted distribution of pacing sites compared with the 19 locations (RV apex, RV outflow tract, and 17 AHA-based sites) evaluated by Arevalo et al. [16]. Our protocol used a subset of at least 5 pacing sites selected from those 19 locations. Because the BZ constitutes the critical substrate for reentry [23,39,40], pacing sites were distributed homogeneously around the MI border, as more distant pacing often elicits redundant reentrant patterns. Consistent with this, a comprehensive AIP applied to the atria showed that electrophysiologically remodeled regions exhibit greater arrhythmia susceptibility and that a reduced number of homogeneously distributed pacing sites can capture up to 90% of induced arrhythmias compared with more exhaustive protocols [60]. Overall, these strategic simplifications enabled high-throughput assessment of the CS-BiV complex while preserving substantial computational scale, yielding reentrant patterns that closely resemble experimental observations (see Sect 3.1).

The selection of S2 coupling intervals was informed by both established literature [16,23,54,61] and the specific electrophysiological constraints of our porcine BiV models. While Arevalo et al. [16] utilized a sequence of premature stimuli, initially delivered 300 ms after the last S1 stimulus with 10 ms decrements, other studies adapting this method to specific porcine or human phenotypes have employed S2 values starting at 250 ms [23,54,61]. In our study, due to HZ/BZ refractoriness, as defined by the Gaur et al. [25] cellular model and its experimentally calibrated tissue properties [20], short S2

intervals or near-MI pacing often failed to capture, preventing VT induction. Longer S2 delays increased inducibility, and variability in such inducibility across S2 values highlighted the value of selecting the shortest effective S2 and identifying the S2 delay that produces the highest inducibility for post-MI remuscularization analyses (G3 and G4 simulations). This establishes a robust framework for assessing inducibility with the Gaur et al. [25] model embedded in BiV simulations.

LAD pigs were consistently more arrhythmogenic than LCx pigs, with each presenting at least one sVT. This finding aligns with clinical observations of a greater incidence of arrhythmia in LAD infarction compared to LCx [62], likely reflecting the larger infarct size and remodeling observed in LAD cases [3]. Our simulations further showed that a larger BZ extent within the infarcted region was associated with greater arrhythmia vulnerability, consistent with prior canine and human studies [39,40]. Exceptions likely arise from BiV modeling constraints, which may introduce artificial low-conduction pathways that sustain reentry (e.g., pig 5) or CS architectures that effectively block reentry (e.g., pig 6).

To our knowledge, this is the first computational study to reproduce the experimentally higher VT burden associated with LAD infarction using biophysically detailed and vessel-specific porcine MI models.

## 4.3 CS as a modulator of post-MI arrhythmia susceptibility

This study identifies the CS as a key determinant of post-MI arrhythmia susceptibility. Although the CS has been recognized as a potential modulator of conduction [63,64], its specific role in post-MI proarrhythmicity remains unexplored. Using anatomically realistic CS distributions (details on construction and validation are provided in [20]), we demonstrate that CS can strongly influence arrhythmic dynamics by creating *electrical bypasses* across ventricular regions. These electrical shortcuts may sustain or terminate reentry or alter MI depolarization timing, facilitating or impairing the phenomenon we term the *capacitive effect*. Although not explicitly labeled as such in other studies, this effect occurs when a region with prolonged APD and slow conductivity (such as the MI or EHT-treated zone) remains activated until the surrounding healthy tissue with a shorter APD recovers excitability. A comparable mechanism was observed by Riebel et al. [59], in which a CS-originated activation slowly traversed the MI, subsequently reexciting the healthy endocardium and propagating retrogradely back into the CS after the refractory period of the HZ had ended.

Our findings align with clinical evidence reporting CS-induced VTs in ischemia, including reentrant bundle branch VT, involving the entire ventricular CS [65], and Purkinje-based intramyocardial VT, primarily involving the left posterior Purkinje fibers [65,66]. Regarding the latter, Bogun et al. [66] identified a plausible Purkinje involvement in post-MI sustained monomorphic VT based on relatively narrow QRS complexes with left or right bundle branch block morphology that meet the following criteria: concealed entrainment at sites exhibiting Purkinje potentials; VT cycle length variability following changes in intervals between Purkinje potentials; or reproduction of the VT morphology through pacing at a site showing Purkinje potentials during sinus rhythm. Garcia-Bustos et al. [63] further confirmed the anatomical substrate for post-MI CS-derived VTs by showing that PMJs are capable of surviving reperfusion-related ischemia in chronic MI pigs.

Computationally, Berenfeld et al. [67] showed that reentries through Purkinje fibers can generate endo-epicardial breakthroughs that sustain polymorphic VT in a non-ischemic canine CS-BiV model, in line with our observations here. Boyle et al. [68] provided additional *in silico* mechanistic evidence of CS-mediated VT modulation. Specifically, they demonstrated that APD prolongation near the PMJs, a phenomenon consistent with our own observations [20], can induce varying local electrophysiological patterns, such as rotor anchoring and wavefront fractionation. More recently, Sayers et al. [69] showed that MI-induced downregulation of fast-conducting connexins and alterations in ion channel expression in non-regenerative CS of postnatal mice slowed conduction and increased the incidence of arrhythmias in a BiV human model with LAD MI. Our study extends and complements these insights by testing CS effects across multiple ischemic substrates and pacing sites, showing how CS critically shapes reentry inducibility and dynamics.

Accurate CS modeling is, therefore, essential to prevent underestimation or overestimation of arrhythmic risk and to ensure the reliability of next-generation *in silico* arrhythmia assessments. Next, more advanced quantitative descriptors [67,70] - including breakthrough counting and spatial localization analysis; rotor-related metrics such as period and

trajectory; phase singularity computation; and quantification of successful versus failed anterograde and retrograde propagation across PMJs - may be systematically incorporated to further elucidate the mechanistic insights introduced herein, while enabling a more rigorous and quantitative assessment of VT complexity by discriminating among anatomically anchored reentry, functional rotor dynamics, and PMJ-mediated propagation mechanisms.

## 4.4 EHT conductivity has a major impact on arrhythmia vulnerability after MI remuscularization

Our simulations demonstrate that the conductivity of the EHT is a key determinant of arrhythmia susceptibility following MI remuscularization. Across the nine porcine-specific BiV models and the two implantation sites, greater and more biomimetic intercellular coupling consistently reduced or prevented increases in arrhythmia vulnerability. These results are consistent with previous computational studies using realistic ventricular representations [17,26,29,30] and models of transmural MI slabs [28,55], where higher EHT conductivity decreased VT burden and repolarization time gradients (a known proarrhythmic marker).

Specifically, Yu et al. [17] similarly reported a reduced VT burden in BiV-MI models when EHT electrical properties approached those of the host ventricular tissue. Fassina et al. [26] found that sVTs disappeared when conductivity reached nearly healthy levels, even in immature repolarization phenotypes. Likewise, our earlier studies showed reduced repolarization gradients with increasing EHT conductivity, irrespective of hiPSC-CM alignment or the degree of graft attachment [29,30].

Taken together, these findings highlight that a mature EHT with high conductivity consistently mitigates, or at least contains, remuscularization-induced VT burden across diverse conditions involving cell alignment, degrees of graft attachment, infarct substrates, and implantation sites.

## 4.5 MI substrate strongly modulates arrhythmia risk after EHT implantation

Following post-MI remuscularization, arrhythmia inducibility exhibited marked artery occlusion dependence, varying substantially in LAD pigs but remaining more stable in LCx pigs. In LAD pigs, the average IS after EHT implantation resembled pre-engraftment values across both implantation sites. In contrast, LCx pigs showed a clear increase in average IS after EHT implantation, although lateral placement reduced reentry compared to central placement. These results extend prior observations by Yu et al. [17], who demonstrated location-specific VT incidence in two MI models where infarcts were confined to the endocardium or were highly diffuse in the epicardium, consistent with our findings in LAD pigs.

In pigs with highly transmural SZ, EHTs created novel isthmuses with fetal-like electrophysiology that, in some pigs (e.g., 4–6), enabled base-to-apex conduction and promoted unidirectional block and reentry. These proarrhythmic effects diminished with increased EHT conductivity, in agreement with previous studies [53]. In some cases, preexisting reentry blocked the EHT conduction path, invalidating any conductivity effects. In pigs where the EHT primarily covered conducting tissue (e.g., 7–12), no additional unidirectional blocks formed, as arrhythmicity was unaffected by conductivity changes. This suggests that inducibility after remuscularization in pigs without transmural SZ depends mainly on EHT-MI spatial relationships and the immature EHT phenotype.

Consistent with Campos et al. [53], we find that low conducting isthmuses are important for the onset of reentry, while reduced conductivity is the main contributor to its sustainability. This mechanism explains why post-remuscularization VT burden in weakly transmural infarcts varies strongly with EHT location at low EHT conductivity but is reduced or contained when EHT conductivity is close to biomimetic values [17]. In contrast, highly transmural infarcts can generate new VTs when covered by the EHT, as shown by Fassina et al. [26] and here. These VTs are suppressed when conductivity is increased.

Overall, our results indicate that the infarct substrate is a primary determinant of VT incidence after remuscularization. For highly transmural SZs, arrhythmicity is reduced when the EHT is implanted laterally to the infarcted region compared to direct implantation over the scar. In contrast, for weakly transmural SZs, VT burden depends on EHT location relative to

the infarct, which emphasizes the need for subject-specific *in silico* models to assess arrhythmic risk and optimize implantation strategies.

### 4.6 Limitations, uncertainties, and future work

**Model characteristics:** The tetrahedralization process employed edge lengths consistent with established literature [16,23,51,71], which have been shown to reliably capture both physiological and arrhythmic dynamics. Specifically, Boyle et al. [72] demonstrated that edge lengths below 400 μm are sufficient to resolve reentrant circuits consistent with experimental observations while mitigating numerical conduction block artifacts. More recently, Bishop et al. [73] highlighted the need for finer discretizations (≤ 200 μm) in low-CV regions to prevent spurious conduction block and reduce variability in arrhythmic outcomes. In our framework, median edge lengths below this 200 μm threshold were achieved within the EHT and adjacent regions. Furthermore, we applied the CV tuning strategies recommended by Bishop et al. [73] for coarser meshes: conductivities in infarcted regions were adjusted to match species-specific OM data obtained under sinus and paced rhythms [20], while settings in healthy and EHT regions produced CVs consistent with experimental benchmarks (see Section 2.6). Although coarser discretizations have been successfully used in prior studies [16,17,26,51,71,72,74], our multiscale meshing approach enhances the robustness of the inducibility assessment by combining high resolution in critical zones with validated conductivity selection.

Low-resolution CMRs [23] and segmentation uncertainties were mitigated using advanced algorithms and extensive experimental validation [20,31]. This supports our finding that LCx pigs displayed more transmural SZs than LAD pigs, likely reflecting both differences in MI induction protocol (proximal LCx versus mid-LAD balloon inflation) and higher out-of-plane LGE-CMR resolution in LAD cases, which allowed thinner BZ layers to be identified [20].

Several parameters were derived from human or canine studies that reproduce the qualitative electrophysiological behaviors typical of large mammalian hearts [16,33,34,45,47,48,50–52]. In our previous work [20], we performed a sensitivity analysis of regional APD ratios, myocardial and CS conductivities, and post-MI $I_{Na}$ remodeling to evaluate the impact of these non-porcine parameters on global electrophysiological characteristics. Variations in APD heterogeneities and conductivities had minimal effects at the tissue and organ levels, whereas small changes in $I_{Na}$ produced substantial alterations in CV within the BZ. In this context, Campos et al. [53] identified CV slowing as a key determinant of reentrant VT probability. These findings highlight the importance of calibrating sodium current remodeling using porcine-specific post-MI data to reduce model uncertainty. Overall, our results indicate that most cross-species parameter choices preserve model robustness, although certain variables may introduce localized bias. While current parameters were calibrated against porcine-specific OM and electrocardiogram data to ensure physiological consistency, future availability of more detailed experimental datasets would enable further refinement and reduce residual cross-species uncertainty.

Accurate CS representation is confirmed to be essential for reducing variability in arrhythmia inducibility. However, precise individualization remains limited, as detailed CS anatomy can typically only be obtained from *ex vivo* histology, and a comprehensive porcine-specific electrophysiological cellular model is still lacking. Currently, endocardial mapping and non-invasive extracellular potential matching represent the most viable methods for validating CS models [51,64,75]. In Rosales et al. [20], we validated the CS architecture used here against porcine subject-specific electrocardiogram and OM data, providing a thorough analysis of the CS architecture's effect on ventricular activation at tissue and organ levels under healthy and MI conditions. The established guidelines were followed for the individualized construction of the CS in all nine models used herein [20]. As additional data become available, porcine-specific functional (e.g., ionic dynamics, conductivity, and anterograde-retrograde PMJ delays) and structural (architecture, PMJ location) CS features should be integrated into the modeling framework to enable more robust preclinical modeling of CS influence on depolarization and arrhythmia. Furthermore, future work should incorporate post-MI region-specific CS conduction delays, recently shown to be highly arrhythmogenic by Sayers et al. [69].

*In silico* AIP: The combination of a limited, low-resolution S2 range and a reduced number of pacing sites, rather than a high-resolution exploration of coupling intervals or a complete 19-site sweep, may fail to capture narrow spatiotemporal windows of vulnerability for VT induction. Consequently, models classified as non-inducible under our protocol could potentially exhibit reentry under a more exhaustive coupling-interval search. Thus, although our *in silico* AIP qualitatively reproduced subject-specific experimental VT patterns with substantially lower computational cost, future studies may benefit from more spatially detailed and temporally extended stimulation strategies. From a spatial standpoint, although no fully automated quantitative criteria were used for pacing-site selection, we implemented a reproducible strategy designed to homogeneously encircle the MI region and assess proarrhythmic BZ heterogeneity [39,40,60]. Future work could incorporate automated methods based on geodesic distance to scar tissue or the identification of slow-conducting isthmuses, as proposed by Campos et al. [54]. Temporally, broader and site-specific S2 coupling interval ranges could be examined using checkpointing strategies and automated estimation of the effective refractory period at each pacing site, following approaches similar to those described by Azzolin et al. [60]. Collectively, these methodological refinements would further standardize the AIP and enable systematic identification of the key spatial and temporal pacing features underlying arrhythmia induction, while maintaining computationally feasibility. Such advances may uncover reentry mechanisms that arise only within narrow temporal windows or specific trigger-substrate configurations.

**Intrinsic heterogeneities of EHTs:** Several intrinsic factors [4,76] may influence outcomes but are simplified here:

- Multicellular composition: Future models could incorporate stromal cardiac cells with fetal-like phenotypes.

- HiPSC-CM variability: Experimental variability in hiPSC-CMs has motivated the development of multiple computational ionic models, resulting in differences in automaticity and susceptibility to arrhythmia [18,77,78]. Recently, Riebel et al. [59] computationally demonstrated that the injection of heterogeneous hiPSC-CMs (ventricular, atrial, nodal) facilitated the generation of post-treatment ectopic activity in a BiV model of LAD MI. Extending our work to incorporate such variability could reveal pig-specific differences in VT burden.

- Cell alignment: HiPSC-CMs often aggregate in islands or exhibit scaffold-driven orientation [10,13]. Here, a disorganized distribution was assumed for hiPSC-CMs in the EHT, consistent with prior slab and BiV simulations showing a limited arrhythmic impact of alignment [17,28,30].

**Extrinsic factors post-implantation of the EHT:** External influences also affect remuscularization outcomes [10,13,17,27,79].

- Degree of attachment: Incomplete physical engraftment may result from air bubbles in poorly adhered constructs, scar tissue from stitching, immune-mediated EHT isolation in xenotransplantation, or shear stress during cardiac contraction-relaxation cycles [10,17]. While our modeling of BiV-EHT coupling as a continuum interface provides a tractable framework for organ-scale VT assessment, representing the discrete, heterogeneous electrical coupling at the graft-host interface remains a key challenge. As noted by Gibbs et al. [80], spatiotemporal heterogeneities in interface coupling can create localized source-sink mismatches that strongly influence focal arrhythmia susceptibility. Prior simulations suggest that homogeneous partial engraftment increases VT risk, whereas heterogeneous partial versus complete engraftment shows comparable risk [17,26]. Consistently, our previous computational study found only minor electrophysiological variations across degrees of heterogeneous engraftment, provided minimal electrical contact existed [29]. Therefore, we focused on complete engraftment in this work.

- EHT-induced ectopy: Experimental studies report a high ectopic burden following cellular injection [7,17,27,79]. Computational evidence indicates that ectopy arises under incomplete engraftment or extreme hiPSC-CM heterogeneity, but not under complete engraftment. Sustained VT further requires a bradycardic beat (or absence of other non-EHT pacing) [59,80] and suppression of CS retrograde conduction [59]. Our results and others [17,26,28–30,55] found no

PLOS Computational Biology

ectopic-driven VT, consistent with the high electrotonic load of the host myocardium suppressing spontaneous activity and the intrinsically slower EHT beating rate, ensuring pacing by the host myocardium. This effect persisted despite using the most recent Paci et al. [46] model, which features enhanced calcium handling and more robust spontaneous beating. Rapid host-induced pacing may further suppress hiPSC-CM automaticity by depleting sarcoplasmic reticulum calcium stores, aligning with findings by Riebel et al. [59]. Recent preclinical and clinical EHT studies [7,8] reporting no arrhythmogenicity support these observations. However, confirmation of stable EHT-myocardium coupling and high cell survival remains necessary. Future studies could investigate bradycardic rhythms and heterogeneous grafts to model conditions more favorable to ectopy.

- Transitional phase: We, like others [17,26], focused on baseline MI and post-EHT implantation endpoints. In this context, our previous work [28–30] showed that the host myocardium dominates both depolarization and repolarization of the EHT once minimal engraftment occurs. Specifically, APD values within the EHT mirrored adjacent myocardial regions. Intrinsic automaticity was suppressed even with the inward, hyperpolarization-activated funny current in the updated Paci et al. [46] model, and CV within the EHT followed that of neighboring myocardium, consistent with prior *in silico* reports [17,26,55]. Nevertheless, dynamic processes during graft integration, such as temporal changes in EHT-myocardium coupling and cell viability (not considered here), may substantially influence arrhythmogenicity. Liu et al. [79] reported early focal VT risk days post-injection, which transitions to relative stability over weeks. Similarly, computational studies show that focal arrhythmia susceptibility waxes and wanes as graft-host coupling gradually increases [80]. For reentrant mechanisms, spatiotemporal variations in electrical coupling may either facilitate or obstruct formation of low-conduction isthmuses (substrates for unidirectional block) and the sustainability of reentrant circuits, resulting in a highly variable VT burden during integration. Future simulations could incorporate time-varying conductivity, stochastic boundary coupling, and hiPSC-CM electrophysiology to capture this high-risk transitional phase between implantation and full electrical integration.

**Outlook:** We developed a realistic EHT-myocardium modeling and simulation framework by integrating an updated hiPSC-CM model with a diverse set of infarct geometries. This approach captures key structural and electrophysiological determinants of arrhythmicity and can be extended as new experimental data emerge. Ultimately, it provides a robust platform to guide case-specific risk assessment and the optimization of post-MI remuscularization strategies.

## 5 Conclusion

We developed a novel computational pipeline to generate porcine BiV subject-specific *in silico* models of infarcted hearts before and after EHT remuscularization, aimed at analyzing VT incidence following therapy. Simulated arrhythmic patterns showed good qualitative agreement with the experimental data. Larger LAD infarcts were linked to higher arrhythmogenicity. CS representation critically influenced post-MI outcomes. Following EHT implantation, VT burden increased in highly transmural scars, particularly when engraftment occurred directly over the infarct; however, it decreased when EHTs were implanted laterally. In less transmural scars, the relationship between VT inducibility and EHT location was highly variable, highlighting functionally similar digital twins as a tool for individualized risk prediction. Finally, biomimetic EHT conductivity emerged as the key factor in mitigating VT burden after remuscularization therapy.

## Supporting information

**S1 Section. Delays of anterograde-retrograde propagation at the PMJ.**
(PDF)

**S1 Fig. Characterization of PMJ anterograde and retrograde propagation delays.** A- Analysis of the postero-apical region (pig 6) following the last S1 stimulus at pacing site 15. Three distinct epicardial breakthroughs are observed, arising

from the interaction between endocardial stimulation and CS depolarization. B- The endocardial stimulus applied at 0 ms (snapshot at 5 ms, top-right) triggers retrograde propagation at a PMJ (CS node ID 2350) at 6 ms (top-left), producing the first epicardial breakthrough (red star, panel A). This wavefront propagates rapidly through the CS (12 ms, bottom-left), culminating in two anterograde PMJ activations at 13 ms (bottom-right, CS node IDs 1592 and 17346), which manifest as the two secondary breakthroughs (black stars, panel A).
(TIFF)

**S2 Fig. Characterization of simulated conduction velocity (CV).** A- Node-wise CV computation was performed on the lateral surface of pig 6 following the last S1 stimulus at pacing site 15. The CV magnitude for a given node was calculated as the mean ratio of Euclidean distance to activation time difference relative to all neighboring nodes within a specified search radius. Sensitivity analysis of the search radius revealed that smaller radii yielded numerical artifacts (infinite values, shown in red) due to neighboring nodes sharing identical activation times (1 ms resolution). CV values in the fast-conducting healthy zone (HZ) stabilized as the radius increased. B- Anatomical segmentation of the lateral ventricular face, highlighting HZ, border zone (BZ), scar zone (SZ), and engineered heart tissue (EHT) regions. C- Median CV values calculated for the HZ, BZ, and EHT across the different search radii tested. A trade-off radius of 2 mm was selected for all subsequent calculations (highlighted in green), as it represented the minimum distance required to ensure numerical stability and convergence across all tissue zones.
(TIFF)

**S1 Table. Results of the arrhythmia inducibility protocol obtained in the G1 simulations.** G1 simulations correspond to the baseline MI-related evaluation. LCx pigs (4–7) are shown at the left table, and LAD pigs (8–12) are shown at the right table. PS: pacing site, NC: no capture, NR: no reentry, nsVT: non-sustained VT, sVT: sustained VT.
(PDF)

**S2 Table. Results of the arrhythmia inducibility protocol obtained in the G2 simulations.** G2 simulations correspond to the assessment of the CS presence. LCx pigs (6–7) are shown at the top of the table, and LAD pigs (11–12) are shown at the bottom of the table. PS: pacing site, NC: no capture, NR: no reentry, nsVT: non-sustained VT, sVT: sustained VT. Note that results in column "with CS" are identical to the outcomes of G1 simulations (S1 Table) where an S2 interval of 295 ms was employed. Additionally, pacing sites which resulted in NC in the G1 simulations were excluded (-).
(PDF)

**S3 Table. Results of the arrhythmia inducibility protocol obtained in the G3 simulations.** G3 simulations correspond to the assessment of the remuscularization and the location of the EHT implantation. LCx pigs (4–7) are shown at the left table, and LAD pigs (8–12) are shown at the right table. PS: pacing site, MI: baseline pre-EHT, L1: EHT at L1, L2: EHT at L2, NC: no capture, NR: no reentry, nsVT: non-sustained VT, sVT: sustained VT. Note that results in column "MI" are identical to the outcomes of G1 simulations (S1 Table) where an S2 interval of 295 ms was employed. Additionally, a low EHT conductivity was used, and pacing sites that resulted in NC in the G1 simulations were excluded (-).
(PDF)

**S4 Table. Results of the arrhythmia inducibility protocol obtained in the G4 simulations.** G4 simulations correspond to the assessment of the EHT conductivity variation when the EHT was implanted in different locations. The left table correspond to the EHT location L1 and the right table correspond to the EHT location L2. LCx pigs (4–7) are shown at the top of the tables, and LAD pigs (8–12) are shown at the bottom of the tables. PS: pacing site, NR: no reentry, nsVT: non-sustained VT, sVT: sustained VT, C10/90: EHT with a conductivity equal to the 10/90% of the value found in the healthy adult-like tissue. Note that results in column "C10" are identical to the outcomes of G3 simulations (S3 Table) where an S2 interval of 295 ms and low EHT conductivity were employed.
(PDF)

**S5 Table. Results of the characterization of the MI substrate.** In the left table, the volumetric ratios between the MI and the BiV, as well as, between the BZ and the MI are presented. In the right table, the epicardial surface ratio of the SZ with whole MI is presented. LCx pigs (4–7) are depicted in the top of the tables while the LAD pigs (8–12) can be found at the bottom of the tables.
(PDF)

**S1 Video. Example results of G1 simulations.** Experimental (top) and simulated (bottom) reentrant activity in LAD pigs.
(MP4)

**S2 Video. Example results of G2 simulations.** Simulated reentrant activity for pigs 6 (top-left), 7 (top-right), 11 (bottom-left), and 12 (bottom-right) with (left) and without (right) the CS.
(MP4)

**S3 Video. Example results of G3 simulations.** Simulated reentrant activity for pigs 4 (top), 5 (middle), and 8 (bottom), previous the EHT implantation (left, MI) and following the central (middle, L1) and lateral (right, L2) to the infarction EHT engraftment.
(MP4)

**S4 Video. Example results of G4 simulations.** Simulated reentrant activity when the EHT conductivity was set to 10% (left) and 90% (right) of that in healthy tissue. Results for pigs 4 (top) and 6 (bottom), when the EHT was implanted lateral (L2) and central (L1) to the infarction, respectively, are shown.
(MP4)

## Acknowledgments

Computations were performed using ICTS NANBIOSIS (HPC Unit at University of Zaragoza).

## Author contributions

**Conceptualization:** Ricardo M. Rosales, Ana Mincholé, Esther Pueyo.

**Data curation:** Ricardo M. Rosales, Gonzalo R. Ríos-Muñoz, Ana María Sánchez de la Nava, María Eugenia Fernández-Santos, Javier Bermejo.

**Formal analysis:** Ricardo M. Rosales, Esther Pueyo.

**Funding acquisition:** Manuel Doblaré, Ana Mincholé, Esther Pueyo.

**Investigation:** Ricardo M. Rosales.

**Methodology:** Ricardo M. Rosales, Manuel Doblaré, Ana Mincholé.

**Project administration:** Esther Pueyo.

**Resources:** Manuel Doblaré, Esther Pueyo.

**Software:** Ricardo M. Rosales.

**Supervision:** Manuel Doblaré, Ana Mincholé, Esther Pueyo.

**Validation:** Ricardo M. Rosales.

**Visualization:** Ricardo M. Rosales.

**Writing – original draft:** Ricardo M. Rosales.

**Writing – review & editing:** Ricardo M. Rosales, Gonzalo R. Ríos-Muñoz, Ana María Sánchez de la Nava, María Eugenia Fernández-Santos, Javier Bermejo, Manuel Doblaré, Ana Mincholé, Esther Pueyo.

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
