## [Decision Letter · Decision Letter 0]

8 Jan 2026

In silico assessment of arrhythmic risk following the implantation of engineered heart tissues in porcine hearts with varying infarct locations

PLOS Computational Biology

Dear Dr. Rosales,

Thank you for submitting your manuscript to PLOS Computational Biology. After careful consideration, we feel that it has merit but does not fully meet PLOS Computational Biology's publication criteria as it currently stands. Therefore, we invite you to submit a revised version of the manuscript that addresses the points raised during the review process. While I have labeled this as a major revision, the reviewers have requested what is more like a fairly large number of modest revisions for clarity.

We look forward to receiving your revised manuscript.

Kind regards,

Andrew D. McCulloch, Ph.D.

Academic Editor

PLOS Computational Biology

Marc Birtwistle

Section Editor

PLOS Computational Biology

**Journal Requirements:**

3) Some material included in your submission may be copyrighted. According to PLOSu2019s copyright policy, authors who use figures or other material (e.g., graphics, clipart, maps) from another author or copyright holder must demonstrate or obtain permission to publish this material under the Creative Commons Attribution 4.0 International (CC BY 4.0) License used by PLOS journals. Please closely review the details of PLOSu2019s copyright requirements here: PLOS Licenses and Copyright. If you need to request permissions from a copyright holder, you may use PLOS's Copyright Content Permission form.

Potential Copyright Issues:

i) Please confirm (a) that you are the photographer of 1D, or (b) provide written permission from the photographer to publish the photo(s) under our CC BY 4.0 license.

4) Please send a completed 'Competing Interests' statement, including any COIs declared by your co-authors. If you have no competing interests to declare, please state "The authors have declared that no competing interests exist". Otherwise please declare all competing interests beginning with the statement "I have read the journal's policy and the authors of this manuscript have the following competing interests"

**Reviewers' comments:**

Reviewer's Responses to Questions

**Comments to the Authors:**

Reviewer #1: This study describes an expansive study of VT susceptibility in the context of EHT implantation using computational models of porcine hearts with MI induced by either LAD or LCx occlusion. The findings indicate that intrinsic maturation of the EHT patch (as represented by increased internal conductivity) and EHT patch location with respect to the scar/BZ are key factors in modulation of VT inducibility. In general, the authors' vivid descriptions of side-by-side simulations (e.g., with or without EHT, or comparing the two EHT locations) provide rich mechanistic insights on factors mitigating or exacerbating the inducibility of reentrant VT in these models. Independent of the main investigational thrust of the paper related to EHTs, the results presented reaffirm the fundamental importance of incorporating a plausible representation of the CS in bi-ventricular models in order to accurately represent the inducibility of reentrant arrhythmia (or the lack thereof) in the context the post-MI landscape. The authors have gone to great pains to exceed the standard of the cardiac EP modeling field in terms of experimental reproducibility by providing links to all relevant source code and repositories containing complete computational models that could plausibly be downloaded and re-used by other research teams with an appropriate amount of effort.

In general, the manuscript is well written and the computational work is exceptionally rigorous. The findings are noteworthy and will move the field forward. The presentation of simulation data alongside experimental findings is exciting. As enumerated below, there are a few major areas (and several other minor points) where I believe the manuscript could be improved, but these are generally intended as constructive feedback and should not be interpreted as a lack of enthusiasm on my part.

Major Feedback:

1) In much of the manuscript, the matching of experimentally observed and simulated reentrant VTs is overstated. The second paragraph of the Discussion section is appropriately calibrated because it highlights the fact that the matching is qualitative in nature, and stresses that there are no experimental data presented corresponding to the critical simulations (i.e., those including EHT patches). Also, the validation study was performed on the simulations conducted in models without the CS, which is surprising given the major importance ascribed to the CS's presence elsewhere in the study. Finally, there were relatively few incidences of sVT in the simulations. Most of the episodes of reentrant arrhythmia observed were nsVT. The authors do not state until late in the Discussion that the majority of arrhythmias observed in optically-mapped hearts were sVT. Despite these issues, the qualitative validation itself is undoubtedly impressive and should be foregrounded accordingly. However, ensure that the findings are properly appreciated by readers, the authors are encouraged to be more explicit about the nature of the validation data (qualitative, limited to simulations of VT in MI models without the CS, differences in sVT/nsVT, etc.) and this information should be reported throughout the paper (esp. in Results).

2) Some rationale should be provided on why the specific S2 values were chosen. In the context of the original paper that motivated this protocol (Arevalo et al. 2016), multiple extrastimuli were given (S2, S3, S4) and the coupling interval preceding each extrastimulus was systematically reduced until the shortest possible while maintining capture. This protocol is not too burdensome to implement computationally since it can be easily achieved by creating a large number of checkpoints after each preceding stimulus, then running a very short stimulus for each possible extrastimulus to assess whether capture occurs. At minimum, this should be discussed as a potential limitation, since in some situations the window of vulnerability for VT initiation might be very narrow (e.g., in some cases presented here, an S2 of 270 or 275 ms might induce reentry in a model otherwise reported as completely non-inducible).

3) The discussion of post-implantation electrophysiological integration of EHTs and ventricular myocardium (esp. regarding ref. 60, doi: 10.1113/JP284244) is somewhat incomplete. A fundamental limitation of continuum models of cardiac EP is that it is challenging to represent heterogeneous coupling at the boundaries between two regions with distinct electrical properties. It would be helpful if the authors could speculate on how heterogeneity in electrical coupling between the patch and the underlying myocardium might affect the dynamics of reentrant VT inducibility, especially given observations from other types of post-MI cell therapy (e.g., predominantly focal VT onset a few days post-injection, then relief from VT risk several weeks after treatment, as reported by Liu et al. doi: 10.1038/nbt.4162)

4) The authors have used a pig ionic model (Gaur et al.) to represent action potential properties in the normal and BZ mycardium, but the Purkinje model (Stewart et al.) was calibrated based on human EP and the EHT properties (Paci et al. 2020) are based on hiPSC-CM. These limitations are justifiable for several reasons (e.g., there is no pig Purkinje model; implantation of pig-based PSC-CM patches is clinically irrelevant for clinical reasons), but should be more clearly presented by the authors (e.g., it should be clearly stated in Methods that the Stewart model is human-based). It would also be helpful for the authors to comment on the potential implications of these inter-species differences in the model components in terms of how their results should be interpreted.

5) No specific rationale is presented for setting the EHT conductivity values to exactly 10% and 90% of the values used in the HZ for the immature and mature EHTs, respectively. The authors should articulate how they arrived at these values. Moreover, it should be emphasized that the functional consequence of randomizing fiber orientation in the EHT regions is that the effective conductivity would be much lower than for organized, laminar myocardium even in the absence of conductivity modification.

Minor Points:

6) When the different rounds of simulations are presented in the Methods, it is initially unclear whether the G1 simulations are run with or without the CS. This becomes obvious when the Results are read, but it would be helpful to specify this more explicitly in Methods.

7) "Thus, this constitutes one of the largest and most complex cardiac electrophysiological simulation studies to date". I'm not sure it's valuable to use superlatives like this. It is certainly a large study of computational cardiac EP compared to some others. Quantitatively gauging the extent to which it is more or less "complex" than others is another matter entirely. I would instead suggest the authors revise this phrasing to correctly emphasize that the computational burden (in terms of compute wall-time required) and complexity of meshes (in terms of finite element average edge length, incorporation of EP heterogeneity) are cutting edge with regards to other comparable studies (e.g., recent organ-scale EP papers focused on hiPSC-CM integration by Yu et al. [2019 Sci Rep, 2022 Cardiovasc Res] and Riebel et al. [2025 preprint, doi: 10.1101/2025.10.03.680241])

8) There are some consistency issues in S1_Video.mp4, in terms of apparent jumps in timing of frame assembly. For instance, in the case shown in the bottom-right corner (Pig 12), there is a discontinuity between the 12- and 13-second mark of the video where the spatiotemporal evolution of Vm(t) changes abruptly in a way that reflects non-physiological behavior. This is likely a issue of how the video was assembled. In general, all videos should be double-checked.

9) The summary data in Fig 5 (left panel) could be improved by adding information about NC sites (i.e., no-capture), which are also relevant in this context. The summary data in Fig 5 (right panel) should be disaggregated so that the individual IS values for each pig model can be appreciated by the reader. As presented, the inter-model heterogeneity is lost. Violin plots or box-and-whisker plots with all individual data points shown could work.

10) The "LI" vs. "HI" subsets of models tested in G2 vs. G1 were somewhat complex for me to grasp. It might be helpful to slightly tweak the way this is expalined in the Methods, and perhaps highlight the corresponding columns in Supplemental Tables 1 and 2 where the identical results are presented. (i.e., for Pig 6, the "with CS" column in Table S2 is identical to the 295 column in Table S1).

11) In videos showing spatiotemporal evolution of Vm(t) in which EHTs are present, it would be very helpful to superimpose a dotted black link delineating the edges of the hPSC-CM region (as in Figs 9-10)

12) The use of the IS convention in the context of Fig 10 is somewhat confusing. Because the analysis is limited to pacing sites in each model that were guaranteed to induce at least nsVT for either L1 or L2 of the EHT, the metric has a different significance than in prior analyses (where inducibility was assessed across multiple pacing sites, many of which did not induce VT). The authors are encouraged to reconsider the use fo the metric in this context, since it artificially inflates the apparent arrhythmogenicity (i.e., values of 1.0 and 0.33 are common here, whereas they are rare in earlier analyses that aggregate data from multiple pacing sites).

13) The discussion of CS as a modulator of arrhythmia inducibility is excellent, but could be further supplemented with discussion of aspects of the VT substrate not included in the present model (e.g., transmural gradients in expression of channels like IKATP), which were examined in earlier generations of computational models incorporating the CS (see doi: 10.1016/j.hrthm.2013.08.010). If anything, these findings along with the new insights elucidated by the authors emphasize the criticality of including the CS in such models if they are to be used to predict clinical inducibility of VT.

14) The authors should carefully consider whether the models can appropriately be described as "digital twins" (i.e., DTs). Use of this terminology is increasingly common, but in some cases it can be misleading and undermine the value of otherwise excellent computational modeling work. Bhagirath et al. (doi: 10.1093/europace/euae295) recently proposed a more nuanced nomenclature (e.g., functionally similar vs. functionally equivalent DTs). The present study lacks some important hallmarks of DTs (e.g., predicting emergent system behavior that the model was not specifically calibrated to reproduce). The absence of optical mapping data from hearts with implanted EHTs to corroborate the simulation predictions also calls into question that aspect of the "twinning". To be clear, none of these issues are critical flaws, nor do they undermine the trustworthiness of the computational work; however, use of the DT terminology remains an issue.

Reviewer #2: Rosales et al. present an in silico study to assess arrhythmogenic risk after post-MI remuscularization with engineered heart tissue (EHT) patches. The authors integrate EHT geometries into nine validated vessel-specific porcine biventricular models (LCx, n=4; LAD, n=5) and evaluate VT vulnerability with an S1–S2 arrhythmia inducibility protocol across pacing sites, coupling intervals, infarct substrates, implantation locations, EHT conductivity (as a proxy for maturation), and the presence/absence of the ventricular conduction system (CS). The authors report (i) higher VT burden in LAD vs LCx infarcts, (ii) strong modulation of inducibility/dynamics by CS “shortcuts,” and (iii) an overall mitigating effect of higher EHT conductivity on VT burden, with implantation location effects depending on scar transmurality. I would like to highlight the well written manuscript and the attempt to connect model behavior to experimentally observed reentrant patterns (optical mapping) as strengths of the study. However, there are some concerns that should be addressed.

Major comments:

Line 28, please add references to support the comment “A promising tissue engineering strategy under active investigation”.

Line 30, the author commented in plural that there are "Recent studies” in plural but there is only one reference given in the manuscript. Please provide more than one reference to support this phrase.

Line 46, while it is true that animal models are the foundation of basic research and different areas of investigation there are several studies in the field of cardiac modeling that use human data. Please add explicitly in the text why using human data could pose limitations and how animal experiments can overcome this limitation. Extend the phrase citing other studies that have been used to study ventricular arrhythmias and ischemia.

Line 61, 73 and along the manuscript, the authors introduce the wording "comprehensive arrhythmia inducibility protocol (AIP). However, during the manuscript I did not understand how the used arrhythmia protocol is “comprehensive”. Please provide sufficient evidence to support why the API is “comprehensive” otherwise I suggest removing it from the manuscript and just call it an “arrhythmia inducibility protocol”.

Line 89, please explain explicitly in the manuscript what is a multi-diffusion problem. Adding the explanation that different Laplace solutions are used to create gradients that are then used to create the different regions of the ventricular myocardium. Additionally, please add to the manuscript the values of the gradients used for each region.

Line 92-93, the conduction system is one of the main parts of the results and analysis of this manuscript. However, few details are given. For example, there are no descriptions about the number of PMJs connections to the myocardium, the resolution of the CS mesh, and the conduction velocity of the CS, etc. Additionally, there is no analysis of the CS construction and the impact on the depolarization, even if this is out of the scope of this study it should be mentioned in the discussion section and add future lines of work in this direction with the data available from the animal models.

Line 98, please add a brief description of which dye was used and the concentration and which motion-blocker was used and at what concentration.

Lines 116-119, please provide more details for reproducibility of the study on “by manually selecting at least 5 of the 19 pacing sites, 118 with the criterion that they were well distributed around the MI”. The reasoning of reducing the computational time could be a valuable reason but this manual selection makes it hard to reproduce the AIP and might benefit future studies on the induction of arrhythmias. Could there be a more reproducible approach to select the 5 pacing sites, such as distance to the scar area which one can think of when looking at figure 2?

Lines 146-148, please add explicitly the use of Meshlab with its corresponding reference. Also add the used method with the corresponding reference using the DOI provided in the Meshlab window of the function of the explicit remeshing method.

Lines 152-156, one of my biggest concerns of this study is the decision of having a bimodal edge length distribution. The authors highlight in the limitation section the importance of the edge length by referencing the study of Bishop et al. However, the authors have not provided sufficient details on the impact of conduction velocity with the bimodal distribution of the edge length. In the EHT area of the model the same value of diffusion coefficient was used which I assume was calibrated for the 381.9 um mean edge length. This will result in a much higher conduction velocity in the EHT area as a mean resolution of 187.2 um. Please provide sufficient evidence that having two areas with very different resolutions will not impact the results of the study and this to the supplementary material of the study as this is a critical point for this work.

Line 176-179, the authors provide diffusion coefficients which are useful but please add explicitly what is the corresponding conduction velocity that was achieved for Gaur et al. model and Paci et al. model. As mentioned before and by the authors the study of Bishop et al. shows the impact of resolution and and the choice of diffusion coefficients should be verified that reaches physiological conduction velocities that will result on numerical artificial block or in this case due to the different resolutions an acceleration of the propagation. Please add explicitly the values of conduction velocities used in this study.

Line 184, the authors mention a “physiological anterograde and retrograde electrical 184 propagation between CS and ventricular tissue”. Please provide the evidence to support this. There should be a delay in the retrograde propagation activation. Was this considered in this work? If this was also considered please add it to the manuscript or comment on the limitation section as this will be a key aspect on the impact of the CS results shown in this study.

Line 191, add explicitly how the insulator was modeled in the monodomain simulations of this study.

Lines 194-199, please add the number of pulses needed for each model to reach the limit cycle. For the Stewart et al. model there is a comment on the automaticity. How about the automaticity of the Paci et al. model? Was there also a stabilization done on the 3D geometry?

Lines 239-240, is this claim valid for this context of cardiac electrophysiology simulation on pigs? Nowadays, most of the studies that use an HPC system are computing millions of seconds of simulated time. Please comment on this claim specially to current studies.

Subsection 3.3, the authors argue that the CS can either sustain VT (via bypasses and “breakthroughs”) or suppress VT, and reports that removing CS decreases IS in HI pigs but increases IS in LI pigs. While this is plausible and aligns with the notion that Purkinje/PMJ pathways can change reentry dynamics, the presented evidence reads mainly as case-based. Please add quantitative descriptors of the mechanism (e.g., number/location of breakthroughs, timing of PMJ activation relative to wavefront arrival, changes in conduction time across regions, rotor period/trajectory statistics, Vm traces, etc) to substantiate the interpretation beyond visual inspection of figures 6 and 7. Additionally, explain the choice of the analysis of figure 8. Why not do the same analysis as in figure 5?

Subsection 3.5, the author shows the results of the states of maturation implemented via changing EHT conductivity and shows that increasing to C90 can suppress VT in some LCx cases (group IS 0.5→0.25) but not LAD (IS stable at 0.57). However, the issue of conduction velocity appears again here. Please check my comments above regarding resolution and conduction velocity. I would suggest changing “Contribution of the ETH maturation” for “Contribution of the ETH electrical coupling” as other aspects of maturation are not explored. As commented before maturation is a more complex aspect that should also affect the APD of the cells in the ETH patch. The authors do not comment on the effect of the Paci et al. model. This model has a long APD90 and automaticity. When coupling the ETH path to the ventricular geometry how are cellular properties changed? How will these different cellular properties impact the suppression of arrhythmias? Was the automaticity of the Paci et al. model suppressed?

Subsection 4.2, how this protocol differs from the Arevalo et al. protocol? This protocol reminds me of an approach called PEERP. Could you please discuss this with your approach?

Line 471, how physiological is your CS? Could you please compare this to other simulation studies that have shown this capacitive effect?

Lines 550-553, how was the conduction velocity to the optical mapping experiments? If I understood it correctly the experiments were under VT, how did the authors calibrate the model using this information? Please add an explicit section on the calibration of conduction velocity from the optical mapping experiments under VT.

Minor comments:

Line 13, fibrotic tissue can contract. Fibrotic tissue does not have an “active” contraction as myocytes but it has “passive” properties which are stiffer than the healthy myocardium but does not mean that fibrotic tissue can not mechanically be deformed. I suggest rephrasing “with non-contractile fibrotic tissue”.

Line 149, I suggest changing the word discretize to tetrahedralized as the surface mesh is already discretized.

Line 170, conductance should be with capital g.

**Have the authors made all data and (if applicable) computational code underlying the findings in their manuscript fully available?**

The PLOS Data policy requires authors to make all data and code underlying the findings described in their manuscript fully available without restriction, with rare exception (please refer to the Data Availability Statement in the manuscript PDF file). The data and code should be provided as part of the manuscript or its supporting information, or deposited to a public repository. For example, in addition to summary statistics, the data points behind means, medians and variance measures should be available. If there are restrictions on publicly sharing data or code —e.g. participant privacy or use of data from a third party—those must be specified.requires authors to make all data and code underlying the findings described in their manuscript fully available without restriction, with rare exception (please refer to the Data Availability Statement in the manuscript PDF file). The data and code should be provided as part of the manuscript or its supporting information, or deposited to a public repository. For example, in addition to summary statistics, the data points behind means, medians and variance measures should be available. If there are restrictions on publicly sharing data or code —e.g. participant privacy or use of data from a third party—those must be specified.requires authors to make all data and code underlying the findings described in their manuscript fully available without restriction, with rare exception (please refer to the Data Availability Statement in the manuscript PDF file). The data and code should be provided as part of the manuscript or its supporting information, or deposited to a public repository. For example, in addition to summary statistics, the data points behind means, medians and variance measures should be available. If there are restrictions on publicly sharing data or code —e.g. participant privacy or use of data from a third party—those must be specified.requires authors to make all data and code underlying the findings described in their manuscript fully available without restriction, with rare exception (please refer to the Data Availability Statement in the manuscript PDF file). The data and code should be provided as part of the manuscript or its supporting information, or deposited to a public repository. For example, in addition to summary statistics, the data points behind means, medians and variance measures should be available. If there are restrictions on publicly sharing data or code —e.g. participant privacy or use of data from a third party—those must be specified.

Reviewer #1: Yes

Reviewer #2: Yes

PLOS authors have the option to publish the peer review history of their article (what does this mean?). If published, this will include your full peer review and any attached files.). If published, this will include your full peer review and any attached files.). If published, this will include your full peer review and any attached files.). If published, this will include your full peer review and any attached files.

...

Reviewer #1: No

Reviewer #2: No

**Figure resubmission:**
---

## [Decision Letter · Decision Letter 1]

23 Mar 2026

Dear Mr Rosales,

We are pleased to inform you that your manuscript 'In silico assessment of arrhythmic risk following the implantation of engineered heart tissues in porcine hearts with varying infarct locations' has been provisionally accepted for publication in PLOS Computational Biology.

Best regards,

Marc Birtwistle

Section Editor

PLOS Computational Biology

Reviewer's Responses to Questions

**Comments to the Authors:**

Reviewer #1: The authors have adequately addressed all the points I raised.

Reviewer #2: I thank the authors for their comprehensive revision and detailed responses to the reviewers’ comments. All of my previous concerns have been adequately addressed, and the manuscript has been improved accordingly. I have no further comments.

**Have the authors made all data and (if applicable) computational code underlying the findings in their manuscript fully available?**

The PLOS Data policy requires authors to make all data and code underlying the findings described in their manuscript fully available without restriction, with rare exception (please refer to the Data Availability Statement in the manuscript PDF file). The data and code should be provided as part of the manuscript or its supporting information, or deposited to a public repository. For example, in addition to summary statistics, the data points behind means, medians and variance measures should be available. If there are restrictions on publicly sharing data or code —e.g. participant privacy or use of data from a third party—those must be specified.requires authors to make all data and code underlying the findings described in their manuscript fully available without restriction, with rare exception (please refer to the Data Availability Statement in the manuscript PDF file). The data and code should be provided as part of the manuscript or its supporting information, or deposited to a public repository. For example, in addition to summary statistics, the data points behind means, medians and variance measures should be available. If there are restrictions on publicly sharing data or code —e.g. participant privacy or use of data from a third party—those must be specified.requires authors to make all data and code underlying the findings described in their manuscript fully available without restriction, with rare exception (please refer to the Data Availability Statement in the manuscript PDF file). The data and code should be provided as part of the manuscript or its supporting information, or deposited to a public repository. For example, in addition to summary statistics, the data points behind means, medians and variance measures should be available. If there are restrictions on publicly sharing data or code —e.g. participant privacy or use of data from a third party—those must be specified.requires authors to make all data and code underlying the findings described in their manuscript fully available without restriction, with rare exception (please refer to the Data Availability Statement in the manuscript PDF file). The data and code should be provided as part of the manuscript or its supporting information, or deposited to a public repository. For example, in addition to summary statistics, the data points behind means, medians and variance measures should be available. If there are restrictions on publicly sharing data or code —e.g. participant privacy or use of data from a third party—those must be specified.

Reviewer #1: Yes

Reviewer #2: Yes

PLOS authors have the option to publish the peer review history of their article (what does this mean?). If published, this will include your full peer review and any attached files.). If published, this will include your full peer review and any attached files.). If published, this will include your full peer review and any attached files.). If published, this will include your full peer review and any attached files.

...

Reviewer #1: No

Reviewer #2: No

---

## [Editor Report · Acceptance letter]

PCOMPBIOL-D-25-02377R1

In silico assessment of arrhythmic risk following the implantation of engineered heart tissues in porcine hearts with varying infarct locations

Dear Dr Rosales,

I am pleased to inform you that your manuscript has been formally accepted for publication in PLOS Computational Biology. Your manuscript is now with our production department and you will be notified of the publication date in due course.

With kind regards,

Anita Estes
